# A molecular vision of fungal cell wall organization by functional genomics and solid-state NMR

Arnab Chakraborty[1,6], Liyanage D. Fernando[1,6], Wenxia Fang [2,6], Malitha C. Dickwella Widanage [1,6], Pingzhen Wei[2], Cheng Jin [2,3], Thierry Fontaine[4], Jean-Paul Latgé[5 ✉] & Tuo Wang [1 ✉]

Vast efforts have been devoted to the development of antifungal drugs targeting the cell wall, but the supramolecular architecture of this carbohydrate-rich composite remains insufficiently understood. Here we compare the cell wall structure of a fungal pathogen *Aspergillus fumigatus* and four mutants depleted of major structural polysaccharides. High-resolution solid-state NMR spectroscopy of intact cells reveals a rigid core formed by chitin, β-1,3-glucan, and α-1,3-glucan, with galactosaminogalactan and galactomannan present in the mobile phase. Gene deletion reshuffles the composition and spatial organization of polysaccharides, with significant changes in their dynamics and water accessibility. The distribution of α-1,3-glucan in chemically isolated and dynamically distinct domains supports its functional diversity. Identification of valines in the alkali-insoluble carbohydrate core suggests a putative function in stabilizing macromolecular complexes. We propose a revised model of cell wall architecture which will improve our understanding of the structural response of fungal pathogens to stresses.

[1] Department of Chemistry, Louisiana State University, Baton Rouge, LA, USA. [2] State Key Laboratory of Non-food Biomass and Enzyme Technology, Guangxi Academy of Sciences, Nanning, China. [3] State Key Laboratory of Mycology, Institute of Microbiology, Chinese Academy of Sciences, Beijing, China. [4] Unité de Biologie et pathogénicité fongiques, INRAE, USC2019, Institut Pasteur, Paris, France. [5] Institute of Molecular biology and Biotechnology (IMBBFORTH), University of Crete, Heraklion, Greece. [6] These authors contributed equally: Arnab Chakraborty, Liyanage D. Fernando, Wenxia Fang, Malitha C. Dickwella Widanage. ✉email: jean-paul.latge@pasteur.fr; tuowang@lsu.edu

Life-threatening fungal infections are found in more than two million people worldwide every year[1–3]. The insufficient efficacy of commercially available drugs, the substantial rise of azole-resistant strains, and the extensive application of immunosuppressive agents call for the development of novel antifungal compounds[4–6]. Polysaccharides in fungal cell walls are absent in humans, making them uniquely suitable as the target of antifungal treatments. A family of drugs (echinocandins) inhibiting the synthesis of β-1,3-glucan, one of the major cell wall components, have been developed and are clinically used[6,7]. The loss of β-1,3-glucan, however, is partially compensated by the increased content of chitin and the paradoxical effect of this drug (reduced activity at high concentration), both of which have restricted the performance of echinocandins[8–10]. To date, among the many cell wall components, only the inhibition of β-1,3 glucan has been successfully involved in the development of antifungals. To enable the identification of new drug targets, it is of high significance to understand the structural dynamics of fungal polysaccharides and their compensatory responses to cell wall stress or injury.

The current study is focused on the cell wall of *Aspergillus fumigatus*, one of the most threatening human opportunistic pathogens and one of the best understood genetically and biochemically[5,11]. The major carbohydrate components of *A. fumigatus* cell walls include chitin, β-1,3-glucan (primarily linear or with β-1,6-branching and β-1,3/1,4-sequences), galacto-mannan (GM), α-1,3-glucan, and galactosaminogalactan (GAG) (Fig. 1a)[2,12]. Until recently the organization of cell walls was only characterized using protocols that require chemical extraction of this polymer network by alkali and other chemicals[13,14]. The chemical method for analyzing the composition of the cell wall sequentially involves enzymatic or chemical degradation, pur-ification of the produced soluble oligomers, and identification of covalent linkages between monosaccharide residues[13]. In addi-tion, these chemical treatments can also separate the poly-saccharides in amorphous alkali-soluble material and fibrillar alkali-insoluble material[12].

Recently, solid-state NMR spectroscopy has been employed to characterize the molecular architecture of cell walls and melanin deposition in multiple fungal species including *A. fumigatus*, *Cryptococcus neoformans*, and *Schizophyllum commune*[15–20]. This nondestructive method allows for the direct use of untreated intact cells and gives atomic-level indications of polymer rigidity and physical packing as defined by the native properties in cell walls[21,22]. The solid-state NMR analysis of *A. fumigatus* cell wall has suggested that α-1,3-glucans are spatially packed with chitin and are likely distributed in a soft and hydrated matrix formed by diversely linked β-glucans[15], whereas the chemical data suggested that α-1,3-glucan and chitin were separated based on their dif-ferential alkali-solubility. Therefore, it is essential to reconcile the NMR-restrained structure and the model based on biochemical analysis[12,15].

For this purpose, we coupled NMR studies with a functional genomics approach using *A. fumigatus* cell wall mutants that have been characterized previously using chemical methods. The par-ental strain is Δ*akuB*[KU80], which is a widely used model strain

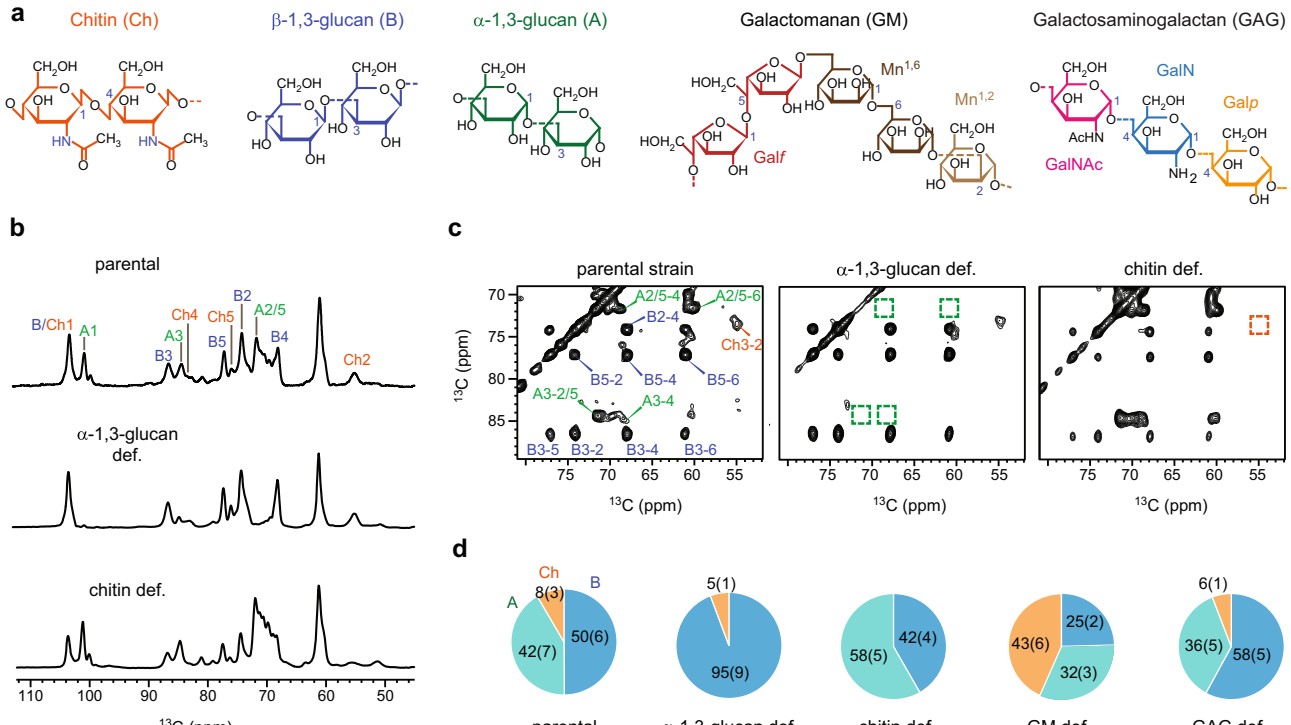

**Fig. 1 Structural changes in the rigid core of *A. fumigatus* mutant cell walls. a** Representative structures of fungal carbohydrates. Abbreviations are shown for different polysaccharides and sugar units. **b** 1D $^{13}$C–CP spectrum showing different intensities for rigid polysaccharides. Abbreviations are used for resonance assignments. For example, A1 denotes α-1,3-glucan carbon 1. Ch and B represent chitin and β-1,3-glucan, respectively. **c** 2D $^{13}$C–$^{13}$C correlation spectrum with 53 ms CORD mixing detecting intramolecular cross peaks. For example, B3-5 is the cross peak between β-1,3-glucan carbon 3 and carbon 5. The missing peaks of α-1,3-glucan and chitin in two mutants are highlighted using dash line boxes. **d** Estimation of polysaccharide composition in the rigid portion of cell walls. Chitin, α-1,3-glucan, and β-glucans are shown in orange, green, and blue, respectively. The percentage values represent the molar fraction of rigid polysaccharides as estimated using the integrals of cross peaks in 2D CORD spectra, which is detailed in Supplementary Table 3. Standard errors included in the parentheses are based on data presented in Supplementary Fig. 3 and computed as described in the Supplementary Methods. Source data of (**d**) are provided as a Source Data file.

resulting from the deletion of the KU80 gene to enhance homologous recombination[23]. Based on this parental strain, four mutants, each of which selectively eliminates a major cell wall polysaccharide, were generated. The first mutant is the quadruple ΔcsmA/csmB/chsF/chsG mutant with the deletion of four chitin synthase genes, resulting in cell walls almost devoid of chitin[24]. The second mutant used is exempt from α-1,3-glucan consecutively to the deletion of genes encoding three synthases (AGS1, AGS2, and AGS3)[25,26]. The third one is a GAG-deficient mutant ensued from the deletion of the GT4C gene encoding a glycosyltransferase, which is devoid of GAG[27]. The fourth mutant has the deletion of KTR4 and KTR7 genes encoding two mannosyltransferases, and this double mutant no longer contains any GM linked to the inner β-1,3-glucan-chitin core, without affecting the N-glycan moiety of proteins[28].

In this work, we employ a series of 2D $^{13}C/^{15}N/^{1}H–^{13}C$ correlation solid-state NMR methods to analyze the uniformly $^{13}C,^{15}N$-labeled hyphal cell walls of the parental A. fumigatus strain and the four mutants described above (Supplementary Table 1). Polysaccharide composition of the rigid and mobile cell wall domains is interrogated in parental and mutant strains. We confirm the functional diversity of α-glucans through its distribution heterogeneity, in the alkali-soluble and alkali-insoluble fractions of the inner and outer cell walls. Our data also show that A. fumigatus substantially reshuffles polysaccharide composition to increase the rigidity and hydrophobicity of cell walls, in response to biosynthesis deficiencies. This study shows the power of a joint genomic, chemical, and biophysical approach to characterize the supramolecular assembly of biopolymers in cell walls and provides a readily applicable method for evaluating the structural responses of fungal cell walls to genetic mutations and external stresses.

## Results

**Polysaccharide structure and a vision of their role in cell wall organization.** For atomic-level characterization using solid-state NMR, we produced uniformly $^{13}C$, $^{15}N$-labeled samples by growing the five strains for 1.5 days in a fully defined medium containing $^{13}C$-glucose and $^{15}N$-NaNO$_3$. Intact cells were directly packed into a solid-state NMR rotor, without any chemical perturbation; therefore, the physical and structural status of the cell wall was kept native. Tailoring the solid-state NMR methods allowed us to selectively detect the rigid and mobile molecules as defined by their native dynamics in cell wall materials, with no relevance to covalent linkage patterns or their susceptibility to chemical extraction (for example, alkali treatment). It is quite common that a single type of polysaccharides could possibly have mobile domains that are present in the soft matrix and rigid phases that are physically packed with stiff molecules such as the cellulose microfibrils in plants and the chitin molecules in fungi[15,29].

The mobile phase represents those molecules with rapid $^{13}C–T_1$ relaxation, which can survive through the short recycle delay used in $^{13}C$ direct polarization (DP). The rigid components described here refer to those polysaccharides that efficiently retain their dipolar couplings and thus can be detected using the dipolar-based $^{1}H$-$^{13}C$ cross polarization (CP) method. These methods have been applied in solid-state NMR studies of carbohydrate-rich materials such as the cell walls of plants and algae as well as the biofilms and cell walls of fungi[18,30–34]. In A. fumigatus, combining these two methods enables efficient detection of both mobile and rigid molecules at ambient temperature (Supplementary Fig. 1). This physical vision differs from the classification accepted after chemical extraction and solubilization of the cell wall, where the alkali-soluble and water-

insoluble molecules are recognized as amorphous polysaccharides as observed by electron microscopy (EM) whereas the alkali-insoluble fraction contains fibrillar molecules also seen by EM[35]. The compositional and dynamical characteristics of all strains were extremely reproducible between different batches (Supplementary Fig. 2).

Only three polysaccharides were found to constitute the rigid core of A. fumigatus cell walls, including chitin, α-1,3-glucan, and β-1,3-glucan (Fig. 1b). The absence of signature peaks confirmed the exclusion of α-1,3-glucan and chitin in the cell walls of their corresponding mutants. The key peaks of α-1,3-glucans, for example, carbon 1 at 101 ppm (A1) and carbon 2/5 at 72 ppm (A2/5), were substantially suppressed in the α-1,3-glucan deficient mutant. Similarly, the resolved peak of chitin carbon 2 (Ch2) at 55.5 ppm, with partial overlap with lipid and protein carbons, was weaker in the chitin-deficient mutant.

Two-dimensional (2D) $^{13}C–^{13}C$ correlation spectra substantially improved the spectral resolution, allowing us to resolve a large number of carbon sites in the rigid macromolecules (Supplementary Fig. 3). The $^{13}C$ full-width at half maximum (FWHM) linewidths are mostly in the range of 0.45–0.75 ppm for the rigid molecules (Supplementary Fig. 4). The chemical shifts are summarized in Supplementary Table 2. Alteration in the polysaccharide amount can be closely examined by tracking the intensities of corresponding cross peaks (Fig. 1c), for instance, α-1,3-glucan carbon 3 to carbon 4 (A3-4) cross peak at (84.5, 69.5) ppm and chitin carbon 3 to carbon 2 (Ch3-2) at (72.9, 55.5 ppm). Analysis of cross peak integrals led to an estimate of the molar fractions of rigid polysaccharides (Fig. 1d). In parental cell walls, the percentages of β-1,3-glucan, α-1,3-glucan, and chitin were estimated to be 50%, 42%, and 8%, respectively (Supplementary Table 3). Defects in α-1,3-glucan biosynthesis were compensated by an upsurge of the β-1,3-glucan amount to 95% whereas the removal of chitin was accompanied by a higher content of α-1,3-glucan (58%).

Although both GAG and GM only exist in the mobile phase (discussed later), the rigid cell wall polysaccharides were still modified in their corresponding mutants (Supplementary Fig. 3). There was no change in the rigid portion of the GAG-deficient cell walls in comparison to the parental strain, however, the mannan deficiency resulted in an increase in the chitin amount (43%), indicative of a concerted change of both rigid and mobile polymers. This increase in the rigid chitin polymer might be associated with the growth defect seen in the Δktr4/Δktr7 double mutant[28]. Two other molecules, α- and β-glucans, exhibited lower amounts in the GM-deficient mutant; therefore, the observed increase of chitin content is not a direct consequence of the reduced amount of GM.

The signals of mobile molecules absent in the above CP-based experiments were preferentially detected using 2D $^{13}C$ DP J-INADEQUATE spectra, which showed numerous carbon peaks from GM, GAG, α-1,3-glucan, and β-1,3-glucan (Fig. 2a). The observed mobile phase has rapid $^{13}C–T_1$ relaxation to survive through the short recycle delay (for example, 2 s) used in $^{13}C$-DP excitation. As this study is using uniformly $^{13}C$-labeled samples and slow magic-angle spinning (MAS) frequencies, $^{13}C–^{13}C$ spin-exchange induces multiexponential relaxation feature, with fast and slow relaxation components for each carbon site[36]. In cell wall NMR, this physical principle has been used to distinguish different domains of mobile molecules present in the soft matrix or in contact and efficient spin exchange with rigid scaffolds, with the former being better detected in the DP J-INADEQUATE experiment[29,31]. The linewidth is typically 0.30-0.75 ppm for dynamic molecules (Supplementary Fig. 4). The variation of linewidths is partially attributable to the highly heterogeneous dynamics of molecules in cellular samples, with the most mobile

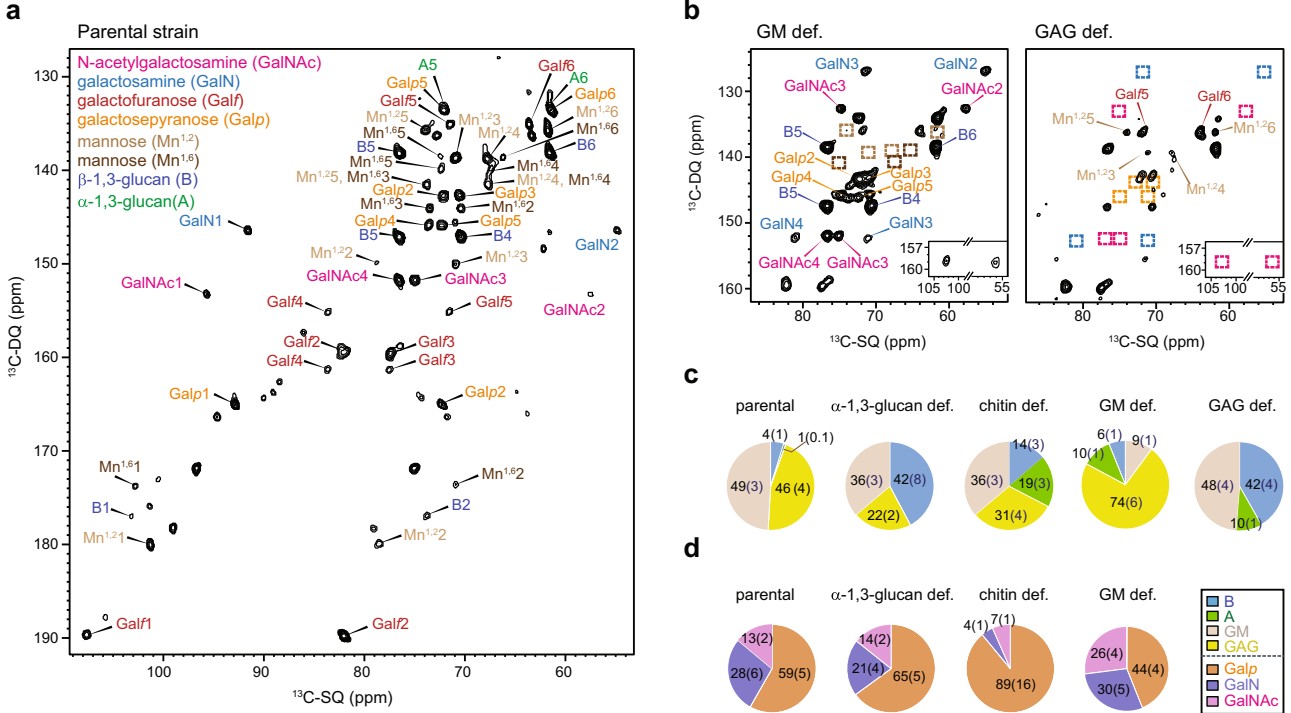

**Fig. 2 The mobile domain of *A. fumigatus* cell wall is rich in GM and GAG. a** [13]C DP J-INADEQUATE spectrum resolving the carbon connectivity for each polysaccharide. Abbreviations are used for resonance assignments and different polysaccharides are color coded. **b** Comparison of mobile polysaccharides in GM- and GAG-deficient mutants. Missing peaks of GM and GAG are highlighted using dash line boxes (typically, brown for mannose, orange for galactopyranose, cyan for galactosamine, and magenta for N-acetylgalactosamine). Insets show the signals of GalN'/GalNAc' residue in the GM-deficient mutant. **c** Molar fractions of mobile polysaccharides in each cell wall sample estimated from peak volume. **d** Monosaccharide compositional changes of GAG observed in the four samples. Standard errors are included in parentheses. The numbers in the pie charts shown by (**c**, **d**) are molar percentages as detailed in Supplementary Table 4. Source data of (**c**, **d**) are provided as a Source Data file.

components showing narrow lines and the partially mobile molecules showing slightly broader peaks where the conformational distribution of a large number of monosaccharide units could not be competently averaged out by motion. Covalent and physical interactions between molecules may also contribute to the observed distribution of NMR linewidth.

Galactomannan (GM) can be tracked using the signals of α-1,2-mannose (Mn$^{1,2}$) and α-1,6-mannose (Mn$^{1,6}$), which showed reduced intensity in the GM-deficient mutant (Fig. 2b and Supplementary Fig. 5). Galactosaminogalactan (GAG) is an exopolysaccharide featured by a complex structure comprised of α-linked galactopyranose (Gal*p*), galactosamine (GalN), and N-acetylgalactosamine (GalNAc) units with no particular order (Fig. 1a)[37–39]. Covalently bonded to a nitrogen, the carbon 2 of GalN and GalNAc exhibited characteristic chemical shifts below 60 ppm, like the carbon 2 signals in chitin. Weak signals have been observed for the GalNAc and GalN carbon 2 at 54-56 ppm. Most of their carbon 1 signals were observed in the range of 92–97 ppm, likely due to the α-linkages and a solvated environment, but weaker signals were also observed at 102 ppm (Fig. 2b, inset). The key signals of GalNAc, GalN, and Gal*p* residues were eliminated in the GAG-deficient mutant.

Differences exist between the two types of galactose units in the mobile polysaccharides. Structurally, both Gal*f* and Gal*p* have 6 carbons, however, the former has a 5-membered ring and a unique large C1 chemical shift at 108 ppm (Fig. 1a). Functionally, Gal*f* is present in the GM of the inner cell wall and in the glycolipids and N- and O- glycans of glycoproteins whereas Gal*p* is present in GAG[39–43].

GM and GAG turned out to be the most populated molecules in the mobile phase of fungal cell walls, each accounting for

46–49 mol% (Fig. 2c). In GM, Gal*f* was the major component, with a moderately different repartition of 1,2- and 1,6-linked mannose residues in all these mutants (Supplementary Table 4). The NMR signals used for screening the polysaccharide composition were provided in Supplementary Table 5. In GM-deficient cell walls, the amount of GM was reduced to 9% but it evaded complete removal (Fig. 2c). The low amount of mannan in this Δ*ktr* mutant could originate from membrane-bound mannan[44] or from the N- or/and O-glycan moieties of the glycoproteins, which, in contrast to the cell wall GM, are untouched in the Δ*ktr* mutants[28]. Moreover, the mannan composition was different: the ratio of Mn:$^{1,2}$ Mn$^{1,6}$ of 5:1 and 1:1 in the parental strain and Δ*ktr4/7* mutant, respectively (Supplementary Table 4), which is in agreement with biochemical data[44,45]. GAG was completely depleted in the Δ*gt4C* mutant, followed by a significant increase in β-1,3-glucan in the mobile region. Mutations regarding the biosynthesis of α-1,3-glucan and chitin, two molecules that are largely rigid, also substantially perturbed the mobile polymers. Therefore, the compositional changes of mobile and rigid molecules happened in a concerted manner. Although NMR showed that Gal*p* was consistently the dominant component (~60–90 mol%) of GAG in most samples, we have observed an almost even distribution of Gal*p*, GalN and GalNAc in the GM-deficient mutant (Fig. 2d). Biochemical data of degraded and isolated oligomers have shown that Gal*p* is a minor component (around 10%) and GalNAc is the major unit[46]. This controversy suggests that Gal*p* plays a key role in the flexibility behavior of GAG observed by ssNMR as has been described for the solubility of GAG in urea in biochemical analyses[39]. Compared to the parental strain, the chitin-deficient mutant also altered the ratio of the three major monosaccharide

residues in GAG (Fig. 2d). When lacking one major polysaccharide, the cell wall does not scale up the remaining polysaccharides proportionally.

Compensatory reactions in response to the lack of a cell wall component due to gene deletion were previously observed with data obtained by chemical and enzymatic analysis[12]. The absence of each of the two rigid polymers, α-1,3-glucan and chitin, led to an increase in the amount of AI-insoluble fraction containing the fibrillar polysaccharides[24,35]. The lack of chitin was compensated by an increase in GAG and the absence of α-1,3-glucan was replaced by β-1,3-glucan. In contrast, the deletion of genes coding for GAG and GM did not modify the ratio of fibrillar and amorphous polysaccharides as distinguished using alkali-treatment[27,28]. Although data obtained by ssNMR or by chemical approaches in the analysis of the cell wall mutants were from different experimental strategies, both methodologies agreed in the fact that the composition of polysaccharides is fully reshuffled to better compensate for structural defects introduced by biosynthesis deficiencies and that no structural rules can be established yet based on these modifications.

**Molecular partitioning after alkali treatment**. For decades, the molecular organization of fungal cell walls and especially of *A. fumigatus* cell walls has been analyzed chemically after solubilization of this water-insoluble matrix by alkali and glycosyl hydrolases[13,47]. We have treated the $^{13}$C/$^{15}$N-labeled parental *A. fumigatus* mycelium with 1 M sodium hydroxide at 65 °C, which discriminates polymers by their alkali-solubility and removes the (glyco-)proteins and (glyco-)lipids bound to the cell wall[13,24]. Previous chemical analysis showed that α-1,3-glucans were only found in the alkali-soluble (AS) fraction, with minor β-1,3-glucan contamination[26]. However, the 1D $^{13}$C CP spectra that selectively detect rigid molecules showed a mixture of chitin, β-1,3-glucan, and α-1,3-glucan in the alkali-insoluble (AI) part (Fig. 3a). The two-phase distribution of α-1,3-glucans was confirmed by their overlapped signals in the 2D $^{13}$C–$^{13}$C correlation spectra collected on both AI and AS fractions (Fig. 3b). Furthermore, α-1,3 glucan was still present in the AI fraction even after a second treatment by NaOH (Supplementary Fig. 6). Intensity analysis showed that α-1,3-glucan accounted for 16 mol% of all rigid polysaccharides in the AI portion of the sample analyzed here, with 57% of β-1,3-glucan and 27% of chitin (Supplementary Table 6). This finding shifts from the prevailing paradigm based on chemical analysis and support the recently reported concept obtained by NMR in which α-1,3-glucan is a structural polysaccharide that tightly packs with chitin to form a hydrophobic and stiff skeleton providing mechanical strengths to the cell walls[15].

When the mobile carbohydrates were specifically analyzed, AI showed peaks from β-1,3-glucan, mannan, chitin, and α-1,3-glucan, while AS exhibited unique signals of α-1,3-glucan and mannan (Fig. 3c). Although the high structural polymorphism of chitin has been recently demonstrated by the statistical analysis of their chemical shifts[16], the presence of chitin in the mobile phase of the AI fraction was still unexpected. These chitin molecules should represent a poorly populated and structurally disordered domain that is associated with matrix polymers, for example, β-1,3-glucans[24]. Both AI and AS samples showed signals from the mannose and Galf residues of GM. Protein signals were mainly observed in the mobile phase of AS molecules (Supplementary Fig. 7), together with a small portion in the rigid phase of AI components as shown later.

After NMR measurements, the same batch of AI and AS samples were subjected to chemical analysis. There was a general agreement between the data computed with the two different methodological approaches: β-1,3-glucans and chitin were the major components of AI and α-1,3-glucan was the dominant component of the AS fraction (Fig. 3d and Supplementary Tables 6, 7). Mannan was distributed in both fractions. The amount of amino acid was low in the AI fraction (2%), where valine is the major amino acid, and increased to 5% in the AS sample. The AI fraction accounted for three-fifth of the total mass of the cell wall and was better populated than the AS part (Fig. 3e). The ratio between β-1,3-glucan and α-1,3-glucan was around 4:1 in the AI fraction but swapped to 1:4 in the AS sample (Fig. 3f), which confirmed the presence of α-1,3-glucan in both AI and AS fractions. Indeed, an earlier study of a mutant lacking the only β-1,3-glucan synthase Fks1 still reported Glc residues in the AI fraction, which may originate from α-1,3-glucans[48].

The only discrepancy happened to GAG, which was detected as a minor molecule (9%) of the AS fraction in chemical analysis (Fig. 3d), but we did not identify its signature peaks (Supplementary Fig. 8) in the temperature range of 280–298 K. This might be resulted from the very limited amount of AS sample due to the low yield in chemical extraction, the potentially unfavorable dynamical scheme, and the low content of GAG in the AS portion. Sensitivity-enhancement techniques, such as Dynamic Nuclear Polarization, might provide a solution to the detection of this polysaccharide and other lowly populated molecules[49–51].

These findings allow us to summarize the partitioning of polysaccharides in four fractions corresponding to the rigid and mobile domains of AI and AS portions (Fig. 3g). β-1,3-glucans span across the rigid and mobile phases of AI fractions while chitin mainly exists in the rigid phase of AI materials. GM remains highly dynamic. The rigid domain of AS portion is dominated by α-1,3-glucan but this molecule also exists in all the other three phases: the distribution heterogeneity is an indicator of its functional complexity.

**Valine, an amino acid associated with the rigid cell wall matrix**. The $^{13}$C DP J-INADEQUATE spectra also showed signals from mobile proteins and only the amino acids showing strong intensities were assigned (Fig. 4a). Some signals of the mobile proteins were retained after alkali extraction, primarily present in the alkali-soluble portion (Supplementary Fig. 7), suggesting that they are polysaccharide-associated proteins instead of the intracellular proteins in transit to be secreted, which are normally removed by the treatment. Protein backbone chemical shifts are sensitive to φ and ψ torsion angles, which is useful for probing the secondary structure[52]. Analysis of the Cα and CO chemical shifts revealed α-helicity of most residues except for tyrosine (Fig. 4b). Protein signals were decreased in the GM-deficient mutant (Fig. 4c and Supplementary Fig. 9). In spite of the variable intensity, the observation of protein signals in the parental and mutant samples suggests that these proteins might be a constitutive component of the cell wall.

We collected 1D $^{15}$N CP spectra to examine the structure of proteins and nitrogenated polysaccharides, primarily chitin (Fig. 4d). GAG was not detected in the CP-based $^{15}$N experiment due to the high mobility of this molecule and the selective detection of rigid components by this technique. Two amide peaks at 124 ppm and 129 ppm, together with an amine peak at 38 ppm, have been resolved. The peak intensity is sample-dependent, revealing major changes in the identities and amount of nitrogenated molecules. Compared to the parental sample, the chitin-deficient mutant showed a decline in the height of the amine signal and the 124-ppm amide peak, which can be attributed to the reduced amount of chitin. In GM- and GAG-deficient samples, the 129 ppm peaks were missing, likely caused by the mobilization or removal of proteins. 2D $^{15}$N–$^{13}$C

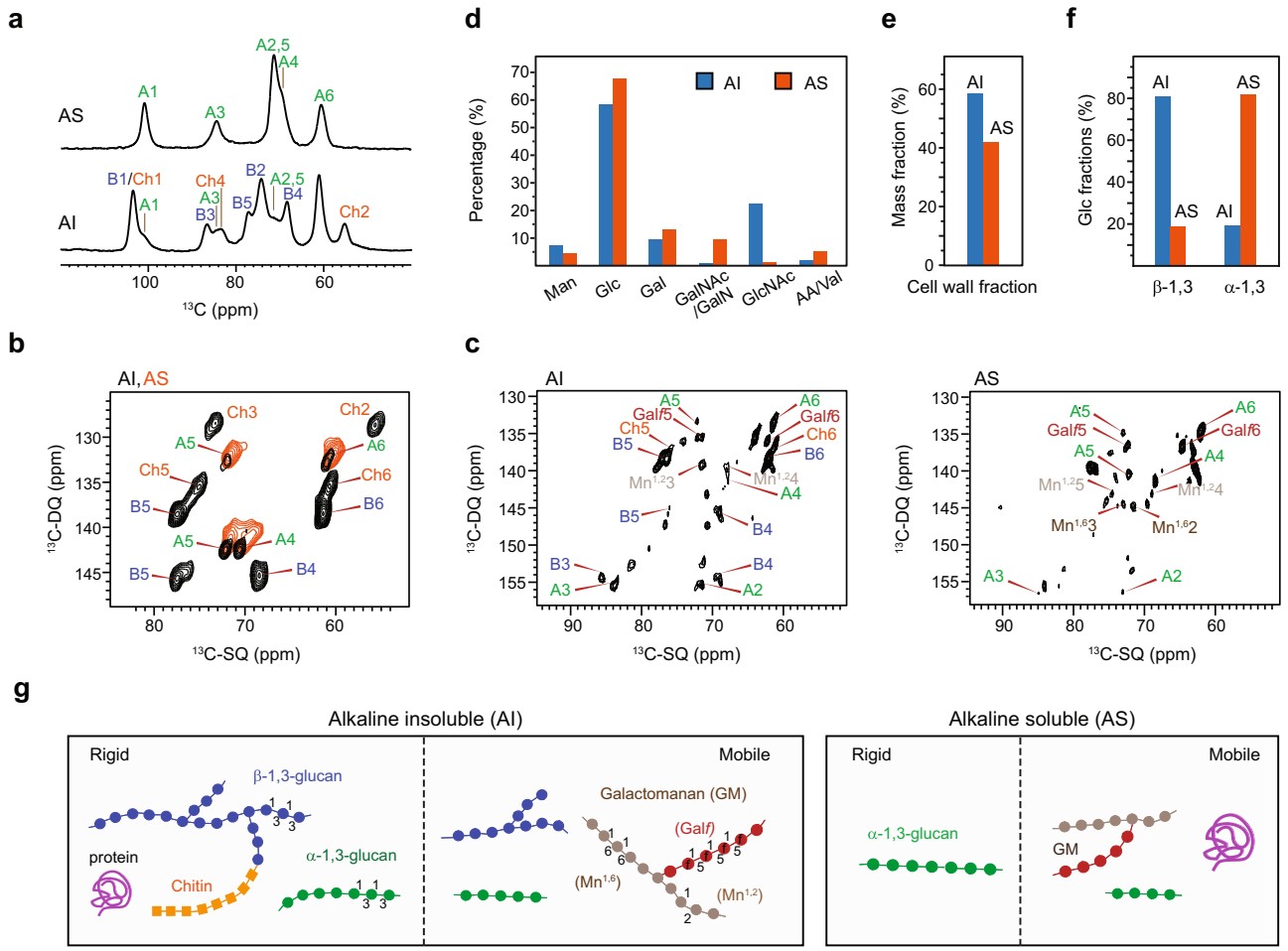

**Fig. 3 α-1,3-glucan is present in both alkali-soluble and insoluble fractions of the parental sample. a** 1D $^{13}$C CP spectra detecting the rigid molecules in the alkali-soluble (AS) and alkali-insoluble (AI) fractions of the parental sample. **b** Overlay of 2D $^{13}$C CP dipolar-INADEQUATE spectra showing the rigid molecules in the alkaline-soluble (AS, orange) and alkaline-insoluble (AI, black) portions of *A. fumigatus* cell walls. Signals of α-1,3-glucans, such as A4, A5, and A6, are present in both portions. **c** $^{13}$C DP J-INADEQUATE spectrum showing the mobile polysaccharides in the alkali insoluble (left) and soluble (right) parts. **d** Compositional analysis of both AI (blue) and AS (orange) fractions obtained by GC-HPLC and enzymatic degradation. The *x*-axis reports different monosaccharide units as well as the amino acids (AA) or valine (Val). **e** Relative mass percentages of the AI and AS fractions. **f** Relative fractions of β-1,3-glucan and α-1,3-glucan. **g** Summary of polysaccharides and proteins identified in both rigid and mobile portions within the alkali insoluble and soluble fractions. Source data of (**d**–**f**) are provided as a Source Data file.

correlation spectra revealed that the 124-ppm signal mainly originated from chitin and 129-ppm peak was from protein backbones (Fig. 4e). The lack of rigid proteins in GAG- and GM-deficient mutants suggested direct associations between structural proteins and these two polysaccharides.

Strikingly, the protein region of 2D $^{15}$N–$^{13}$C spectra mainly has valine (V) signals (Fig. 4e), with only minor contributions from other hydrophobic amino acids such as leucine and serine. This unexpected finding was verified by the strong valine cross peaks observed in the aliphatic region of the 2D $^{13}$C-$^{13}$C correlation spectrum (Fig. 4f). The same signals were fully retained in the alkali-insoluble carbohydrate core of the cell wall, which mainly contains the covalently linked mannan-β-1,3-glucan-chitin complex, but disappeared in the alkali-soluble fraction (Fig. 4g). Further confirmation is provided by the chemical analysis after acid hydrolysis, where the 2% of amino acid content in AI is predominantly valine (Fig. 3d). The presence of rigid valine may suggest a structural function in polysaccharide complex that has never been investigated.

**Polymer dynamics and hydration in the mutants**. The dynamical and hydration characteristics of biopolymers reflect the extent of molecular aggregation and water permeability, which helps to rationalize the structural organization of cell walls. Enhanced rigidity and hydrophobicity are typical indicators of large or ordered aggregates, for example, the chitin and cellulose microfibrils in fungi and plants, respectively[21]. In contrast, molecules spatially separated from these mechanical cores are typically mobile and hydrated. The motional dynamics of polysaccharides on the nanosecond timescale were probed using $^{13}$C spin-lattice ($T_1$) relaxation, which was measured as an array of 2D $^{13}$C–$^{13}$C correlation spectra with a variable z-filter (Supplementary Fig. 10 and Table 8). The use of CP selected the rigid components that are structurally meaningful. Due to the perturbation of the spin-exchange effect[36], the experiments and the use of uniformly labeled materials do not allow the accurate determination of the $^{13}$C–$T_1$ relaxation. Therefore, we only use it as a qualitative indicator of polymer dynamics. After 1 s of relaxation, the signals of β-1,3-glucans decayed rapidly, but the α-1,3-glucan cross peaks still retained high intensities (Fig. 5a). Therefore, polymer dynamics are heterogeneous. The data were fit using single exponential equations to obtain $^{13}$C–$T_1$ relaxation time constants for different carbon sites (Fig. 5b). In the parental sample, the average $^{13}$C–$T_1$ relaxation times for β-1,3-glucan, α-

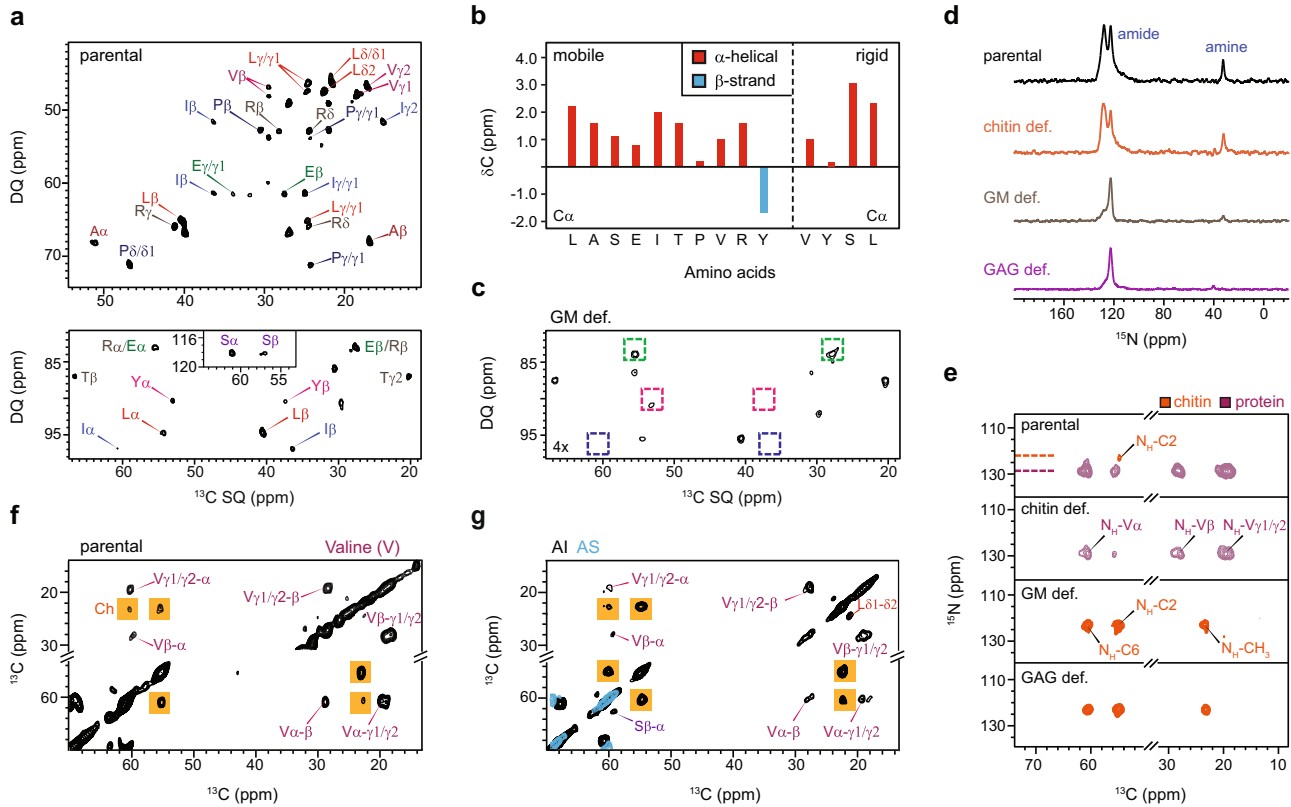

**Fig. 4 Structural assembly of glycoproteins in fungal cell walls. a** 2D $^{13}$C DP J-INADEQUATE spectrum detecting the amino acids of mobile proteins. The assignments represent the amino acid types and carbon sites. For example, $A_\beta$ represents the carbon-β of alanine. The inset shows the signals of Serine. **b** Backbone $^{13}$C chemical shifts suggest the dominance of α-helix secondary structure in both mobile and rigid phases of proteins. **c** Removal of galactomannan results in protein depletion indicated by the decline in amino acid signals as highlighted using dash line boxes. **d** 1D $^{15}$N CP spectra showing multiple amide and amine signals from cell wall polysaccharides and proteins. The $^{15}$N signals vary in the parental sample and mutants. **e** 2D $^{15}$N–$^{13}$C correlation spectra showing chitin (orange) and protein (purple) signals. Chitin signals are missing in the chitin-deficient mutant. Rigid proteins are absent in the GM-deficient and GAG-deficient samples. **f** Valine is the major rigid amino acid in the whole cell of the parental sample as shown by the 2D $^{13}$C–$^{13}$C CORD spectrum. Chitin signals are shown in yellow boxes. **g** Valine is preserved in the rigid portion of the alkali-insoluble (AI) part but becomes absent in the alkali-soluble (AS) fraction. Source data of (**b**) are provided as a Source Data file.

1,3-glucan, and chitin were 1.2 s, 3.3 s, and 2.1 s, respectively. Therefore, the rapid local reorientation is most pronounced in β-1,3-glucan, and becomes subsequently less in chitin and even less in α-1,3-glucan. When α-1,3-glucan was removed, chitin became further rigidified as evidenced by its 3.5 s average $^{13}$C–$T_1$ but β-1,3-glucan became even more mobile. This is an indicator of polymer separation in the α-1,3-glucan-deficient cell walls, as the spin exchange, an effect averaging the $^{13}$C–$T_1$ of closely packed molecules, failed to equilibrate between the two polysaccharides. However, in chitin-deficient cell walls, both α-1,3- and β-1,3-glucans became less dynamic, with their time constants increased to 4.1 s and 2.1 s, respectively.

We further conducted a water-to-polysaccharide $^1$H polarization transfer experiment to examine the changes brought about by genetic mutation to the water accessibility of polysaccharides[53,54]. This experiment depends on a $^1$H-$T_2$ relaxation filter to eliminate all polysaccharide magnetization and then transfers the water $^1$H polarization to carbohydrates so that only carbohydrates with bound water can be detected. The time taken for the signal to reach equilibrium can be used as an indicator of water retention around each carbon site (Supplementary Fig. 11 and Table 9). The polysaccharides with a slower intensity buildup have a reduction in water accessibility (Fig. 5c).

In the parental strain, chitin and α-1,3-glucan require long buildup times due to the formation of a rigid and hydrophobic complex by these two polysaccharides (Fig. 5d). In contrast, the

mobile β-1,3-glucan has short buildup time constants, thus forming a soft and hydrated matrix. Compared to the parental sample, all mutant cell walls are more hydrophobic. Polysaccharides have a compromised capability of retaining water molecules in these mutants. This effect is probably caused by the formation of denser cell walls in these mutants, and consequently, enhanced molecular aggregation, which might serve as a mechanism of fungal cell wall remodeling in response to structural defects or external stresses.

## Discussion

High-resolution solid-state NMR data and chemical analysis of the intact cells and alkali-treated materials of *A. fumigatus* have substantiated our understanding of fungal cell wall architecture. To the best of our knowledge, the physical vision of polymer mobility and the chemical perspective of alkali-solubility have never been combined before. A structural scheme is constructed to represent the conceptual setup of the parental cell wall, which is composed of an outer shell and an internal domain (Fig. 6a). A mobile layer containing GM, protein, a small amount of α-1,3-glucan, and GAG should be mainly at the outside of the cell wall. The occurrence of these molecules in the external position has been shown by immunolabelling with specific antibodies[47,55,56], and NMR data revealed their dynamic nature. Part of the GM molecule is dissolvable in alkali while the part covalently bound to the chitin-glucan complex remains insoluble despite its high

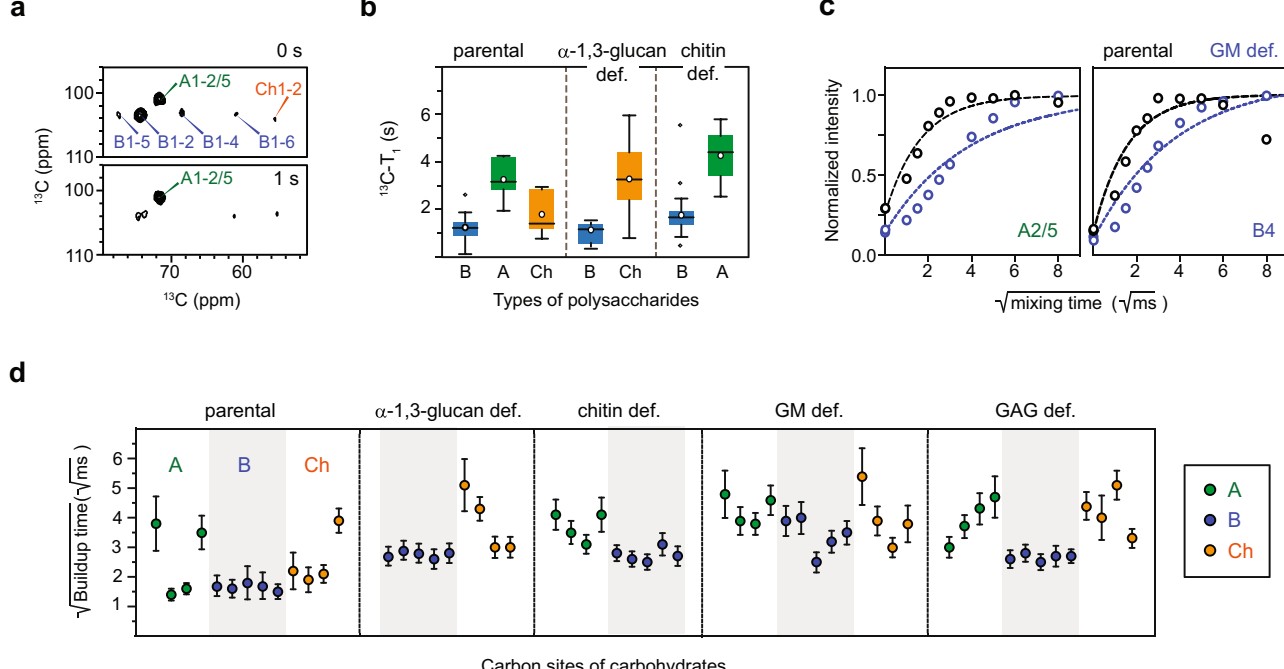

**Fig. 5 Modulated dynamics and water contact of polysaccharides in *A. fumigatus* mutants.** The NMR data of β-1,3-glucan (B), α-1,3-glucan (A), and chitin (Ch) are plotted in blue, green, and orange, respectively. **a** Representative 2D $^{13}$C-$^{13}$C spectra with 0 s (top) and 1 s (bottom) z-filter time for measuring $^{13}$C-$T_1$ relaxation. Signals of α-1,3-glucans are effectively retained after 1 s, indicating the slow $^{13}$C-$T_1$ relaxation of this polysaccharide. **b** Box and whisker diagram plotting the $^{13}$C $T_1$ relaxation time constants of β-1,3-glucan (B; $n = 19$), α-1,3-glucan (A; $n = 6$), and chitin (Ch; $n = 13$) for parental and mutant samples. The box starts from lower quartile to upper quartile, with the horizontal bar and the open circle presenting the median and mean, respectively. The length of the whiskers is determined by the product of 1.5 and interquartile range, with outliers shown as separate dots. **c** Representative buildup curves for the water-edited spectrum of parental (black) and GM-deficient (blue) samples. **d** Data representing the buildup time constants that reflect the degree of water retention at various carbon sites of different polysaccharides. Time constants for β-1,3-glucan ($n = 5$), α-1,3-glucan ($n = 4$) and chitin ($n = 4$) are generated from the fit to exponential function. Each point reflects the best-fit value for buildup time constant ± s.e. Shaded area represents the data of β-1,3-glucan. The data plotted in (**b**, **d**) are summarized in Supplementary Tables 8, 9, respectively.

mobility. Our results also confirmed a recently proposed structural scheme[15], where the inner domain is comprised of a stiff and hydrophobic complex of α-1,3-glucan and chitin, which is distributed in a soft and hydrated matrix of β-glucans. Chitin and β-glucans are joined together by covalent linkages, forming the rigid mechanical hotspots that are resistant to hot alkali treatment[26,35], whereas α-1,3-glucan is physically associated with the chitin-β-glucan-GM core as shown by NMR data. The unequilibrated dynamics suggests that some β-glucans and α-1,3-glucans remain distant from chitin; the former are mobile and alkali-insoluble while the latter are rigid but extractable. Therefore, there is no direct correlation between the chemical digestibility and the rigidity of a molecule.

These studies have shown the synergism of both chemical and biophysical methods. SsNMR has identified a prominent role of α-1,3 glucans in the cell wall structuration whereas chemical analysis has often missed the presence of this polysaccharide in the alkali-insoluble fraction. Similarly, the presence of β-1,3-glucans in the alkali-soluble fraction was underestimated. Chemical analysis may be more accurate to identify the presence of a polysaccharide in a very low concentration: this could be one of the reasons for not seeing the GAG signals in the AS fraction or chitin signals in the chitin-deficient mutants.

Earlier chemical analyses have shown that the composition varies between mycelium and conidium cell wall and the culture medium used[12,57], which could be at the origin of the discrepancies seen between our earlier ssNMR study[15] and the present one. Previously, 1,6- and 1,4-linkages were identified in the β-glucans[15], with the former likely attributable to the

branching points of β-1,3/1,6-glucan and the latter belonging to β-1,3/1,4-glucans. However, such signals were not detected in the current samples. The wild-type strain (RL 578) used in the previous study[15] differs from the one for the current study. In addition, the fungal material used previously was obtained after 14 days of growth in unshaken conditions in a sucrose-based medium. Under these experimental conditions, the material recovered was somehow heterogenous with conidium and mycelium and autolyzed mycelium due to the long growth time. In the current study we use short culture times to recover actively growing mycelium and in a shaken condition to recover a homogenous mycelial pellet, which is a well-controlled system for analyzing the cell wall.

Comparing the parental and mutant cell walls has made it possible to evaluate the structural role of each polysaccharide. Removal of either α-1,3-glucan, GM, or GAG will result in a moderate decline in the average thickness of cell walls (Supplementary Fig. 12), suggesting that the overall biosynthesis of cell wall component has been quantitatively reduced. In the α-1,3-glucan-deficient cell walls, *A. fumigatus* tunes up the synthesis of β-1,3-glucan in the inner core (Fig. 1d). Without α-1,3-glucan as spacers, chitin polymers now become tightly packed as depicted in Fig. 6b, which explains the enhanced rigidity and reduced water accessibility of chitin in this mutant (Fig. 5b, d). The structural roles of chitin and α-1,3-glucan are not interchangeable: the removal of most α-1,3-glucan is not associated to growth defects since the α-1,3-glucan-less mutant is growing like wild-type strains. There is still a missing link between the molecular rigidity and assembly with the mechanics and growth as shown in

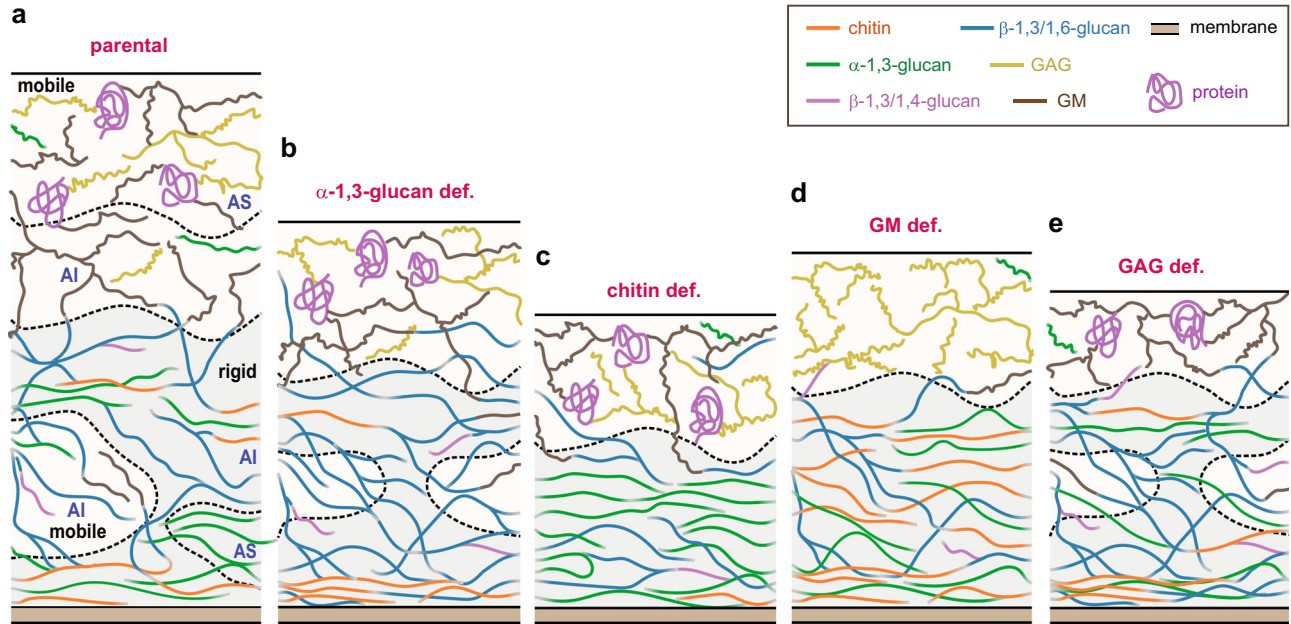

**Fig. 6 Structural scheme of fungal cell walls substantiated by NMR data and mutant strains.** For each sample, the mobile and rigid phases are highlighted in pale yellow and pale blue, respectively. **a** Cell walls of the parental sample, with the alkali-soluble (AS) and alkali-insoluble (AI) portions labeled. The rigid and mobile portions of AI and AS are also shown. The molar fractions of the rigid and mobile domains from solid-state NMR have been considered, but the scheme may not be strictly to scale. The molecule types are labeled and color coded. Templated from the parental cell wall, schematic illustrations are also shown for the four mutants devoid of (**b**) α-1,3-glucan, (**c**) chitin, (**d**) GM, and (**e**) GAG, with the major changes shown. The cell wall thickness is proportional to the average value of each strain observed by TEM, but a broad distribution of thicknesses was observed in Supplementary Fig. 12.

plants[58,59]. Chitin bears a variety of hydrogen bonds using its amide and carbonyl groups[16], which increases the entropy of the system and thermodynamically stabilizes the rigid phase. Hence it is not surprising that the chitin-deficient mutant showed morphological defects, likely due to the failure of cell walls to withstand high turgor pressure during cell growth. On the molecular level, the inner domain should be predominantly a binary mixture of α- and β-1,3-glucans (Fig. 6c). It is likely that these two molecules are extensively associated, which, together with the increase of α-1,3-glucan content, could explain the increased rigidity and hydrophobicity of both molecules (Fig. 5). The chitin-β-glucan-GM core is no longer present in the chitin-deficient mutant. It also becomes questionable whether the inner domain still contains mobile mannan and glucans. GM deficiency depletes proteins but increases the content of GAG. This agrees with the similar roles played by these two mobile molecules present in the outer cell wall layer. Simultaneously, we have observed a fivefold upsurge in the chitin content, and consequently, the production of an extremely hydrophobic and rigid cell wall (Fig. 6d), which is speculated to be a compensatory effect to the loss of cell wall mannan. Supporting this hypothesis, the GAG-deficient cell wall cannot retain water molecules, although its inner domain only shows minor compositional changes when compared with the parental strain (Fig. 6e).

Fungi are adapting two structural principles to respond to cell wall defects. First, the impaired biosynthesis of any polysaccharide will be compensated by compositional changes in both the internal and external domains[23–28]. However, the high level of complexity in these compensatory mechanisms in response to cell wall stress suggested a multitude of coordinated and interacting biosynthetic pathways more complicated than early thought. Second, the re-structuring cell wall tends to increase the polymer rigidity but decrease the water retention in the mesh of the inner domain (Fig. 5d). The balance of plasticity and rigidity

maintained in parental strain has been changed in the mutants, thus perturbing the dual functions of cell walls in maintaining cellular integrity and accommodating cell growth. We suspect that these rules allow fungi to resist not only mechanical deficiencies but also environmental stimuli.

We have observed the coexistence of a significant amount of valine residues with polysaccharides in the parental cell walls (Fig. 4f, g), which becomes undetectable in GM- and GAG-deficient mutants. The origin of the valine is unknown. If present as peptides, they could come from the GPI signal domains that are rich in valine and are removed from the GPI-anchored proteins present in high amount in the cell wall. However, no data showed to date the involvement of this signal peptide after its release from the linkage of the protein moiety to the GPI anchor[60,61]. The unexpected results have however suggested a role of valine residues in the association between cell wall proteins and glycans in *A. fumigatus* and suggested the function of GM and GAG in stabilizing proteins on the cell wall surface. The occurrence or modification of the protein outer layer has not been investigated by SDS-PAGE in these different cell wall mutants. The coexistence of the polysaccharide-protein complexes on the cell wall surface may have impacts on the immune recognition of the fungus by C-type lectins. Such findings certainly deserve further investigation and the isolation and characterization of valine-rich fractions from cell walls.

This joint comparative study of the cell wall structure using two complementary biophysical and chemical approaches has paved the way for future exciting research avenues[20,62,63]. Our results have better revealed the great plasticity of the fungal cell wall and the capacity of the fungus to implement different strategies to survive in the case of the absence or significant modification of an essential cell wall component. This research strategy may reveal new compensatory pathways which could explain why the absence of α-1,3-glucans did not modify fungal

growth, and at the opposite, the absence of β-1,3-glucan or chitin or mannan had a strong morphological impact. These are major reasons for the difficulty to setup an antifungal strategy that targets the cell wall[64,65]. A substantiated molecular understanding of how cell walls structurally respond to antifungal treatments or mutants may guide the design of new antifungal compounds to combat invasive infections.

## Methods

**Preparation of isotopically labeled samples**. To obtain isotopically labeled fungal cells, minimum media containing $^{13}$C-glucose as the sole carbon source and $^{15}$N-NaNO$_3$ as the sole nitrogen source were prepared, with the detailed composition listed in the Supplementary Methods. The parental strain used in this study was $\Delta akuB^{KU80}$, a widely used model strain[23]. The α-1,3-glucan deficient strain was the triple mutant with Ags1p, Ags2p, and Ags3p encoding genes deleted[26]. The chitin-deficient strain was the quadruple $\Delta csmA/csmB/chsF/chsG$ mutant in which the genes encoding both chitin synthase family 1 ($csmA$, $csmB$, and $chsF$) and family 2 ($chsG$) were deleted[24]. The GM-deficient strain was the double knockout mutant of $KTR4$ and $KTR7$, encoding two KTR mannosyltransferases[28]. The GAG-deficient strain was the knockout mutant of $gt4c$ that encoding GAG synthase[27]. Conidia of $5 \times 10^8$ from the parental and the four deficient mutants were inoculated into 100 mL $^{13}$C,$^{15}$N-labeled media at 37 °C under 200 rpm for 36 h growth. The mycelia were harvested by filtering through two layers of miracloth, and then washed extensively using ddH$_2$O. Around 100 mg of the never-dried and intact mycelia of the parental and mutant strains were used for solid-state NMR structural characterization. Three batches of replicates were prepared for each strain under identical conditions. The NMR fingerprints of all strains were highly reproducible between batches (Supplementary Fig. 2).

**Alkali treatment and sugar analysis**. Alkali treatment was conducted on the parental sample. After flash frozen in liquid nitrogen, the mycelia were stored at −80 °C for further manipulations. Cell wall extraction and alkali fractionation were proceeded[66]. Briefly, mycelia were ground and added into 50 mL tubes. To get rid of cell wall-bound proteins mixtures of 50 mM Tris, 50 mM EDTA, 2% SDS and 1 mM TCEP were added and boiled for 20 min twice. After removing the supernatant, cell wall pellets were washed 6 times and lyophilized. Alkali fractionation was carried out with 1 M NaOH in 0.5 M NaBH$_4$ for incubation at 68 °C for 1 h. After centrifugation, the supernatant was the alkali-soluble fraction, which was dialyzed in ddH$_2$O for 2 days. The alkali-insoluble pellet was thoroughly washed by ddH$_2$O until the pH reaches 6. The AS and AI fractions were lyophilized and rehydrated for solid-state NMR studies. The amount of sample recovered was around 15 mg of the AI sample and 10 mg of AS fraction was obtained for the batch analyzed. Hexosamines were identified and quantified by high-performance anion exchange chromatography (HPAEC) on a CarboPAC-PA1 column (Dionex) after acid hydrolysis (8 N HCl, 4 h at 100 °C) using glucosamine and galactosamine as standards. Monosaccharides were analyzed by gas liquid chromatography as their alditol acetates obtained after hydrolysis (4 N trifluoroacetic acid, 100 °C, 6 h) followed by reduction with sodium borohydride and peracetylation[67]. Degradation of α- and β-1,3-glucans of the cell wall fractions was undertaken with recombinant α-1,3-glucanase from *Trichoderma harzianum* and recombinant β-1,3-glucananses from *Thermotoga neapolitana*. Digestions were undertaken by treating the cell wall fraction with enzyme solution in 50 mm sodium acetate buffer for up to 96 h at 37 °C. Degradation products were analyzed by HPAEC[68,69]. Amino acids were classically identified after 24 h 6M HCl hydrolysis and ninhydrin derivatization before quantification[70].

**Solid-state NMR experiments**. Most of the solid-state NMR experiments were conducted on a Bruker Avance 800 MHz (18.8 Tesla) spectrometer except for the measurements of $^{13}$C–T$_1$ relaxation and $^{13}$C–$^{13}$C radio frequency-driven recoupling (RFDR) experiments, which were performed on a Bruker 400 MHz (9.4 Tesla) NMR. The radio-frequency field strengths were 62.5–83.3 kHz for $^1$H hard pulses and CP, 50-62.5 kHz for $^{13}$C, and 41.5 kHz for $^{15}$N in all experiments unless specifically mentioned. The $^{13}$C chemical shifts were reported on the tetra-methylsilane (TMS) scale and externally referenced to the Met Cδ of a model peptide N-formyl-Met-Leu-Phe-OH (f-MLF) at 14.0 ppm[71]. The analysis and plotting of NMR data were achieved using TopSpin, Microsoft Excel, OriginPro, and Adobe Illustrator.

Resonance assignments of cell wall biomolecules were made using: (1) 2D $^{13}$C–$^{13}$C correlation spectra with $^{13}$C–CP and a 53 ms CORD mixing for rigid components[72,73], (2) 2D refocused J-INADEQUATE spectrum with $^{13}$C–DP and recycle delay of 2 s for the mobile components[74,75], (3) 2D SPC-5 dipolar-INADEQUATE spectrum with $^{13}$C CP for the rigid components[76], (4) 2D $^{13}$C–$^{15}$N NCA(CX) heteronuclear correlation spectrum with a 5 ms $^{15}$N–$^{13}$C CP and a 15 ms or 100 ms PDSD mixing time for nitrogenated molecules[77], and (5) 2D $^{13}$C–$^{13}$C radio frequency-driven recoupling (RFDR) experiment for the selective detection of one bond cross peaks with a recoupling time of 1.5 ms[78]. Most data were collected at 298 K under 10 kHz MAS; only the SPC-5 dipolar-

INADEQUATE experiment was conducted at a slow MAS of 7.5 kHz. Experiments 1, 2, 4 were collected using the intact cells of parental *A. fumigatus* and mutants. Experiments 3 and 5 were conducted on the alkali-soluble and insoluble samples. Chemical shifts previously obtained on model polysaccharides or cell wall materials were used as references for assigning the signals[79]. The experimental parameters of all NMR experiments were provided in Supplementary Table 1.

Compositional analysis of the polysaccharides by solid-state NMR in the rigid and mobile portions of cell walls was achieved by taking the integrals of well-resolved cross peaks in 2D $^{13}$C–$^{13}$C correlation spectra: the 53 ms CORD spectra for rigid components and 2D refocused $^{13}$C DP J-INADEQUATE spectra with 2 s recycle delays for the mobile components. For the rigid molecules, the results of well-resolved C1–C3, C1–C2, and C1–C4 cross peaks were averaged. For the mobile polymers, the average of the resolved C1–C2 spin connections gave their relative amount. A more detailed description of the compositional analysis and error propagation is included in the Supplementary Methods. The best-fit relaxation time constants are plotted as a box and whisker diagram.

A series of 1D water-buildup curves were measured using a $^1$H-T$_2$ relaxation filter of 0.6 ms × 2, which abolished 90% of carbohydrate magnetization but retained 80% of water magnetization. A $^1$H mixing period varied from 0.1 μs to 64 ms is then used to allow water-to-polysaccharide polarization transfer, followed by a $^1$H–$^{13}$C CP to enable high-resolution $^{13}$C detection[54,80]. The buildup curves of intensities were plotted for each resolvable carbon site.

The dynamics of polysaccharides in the parental and mutant samples were probed using NMR relaxation. $^{13}$C–T$_1$ relaxation was measured at 298 K under 10 kHz MAS on a 400 MHz spectrometer to provide information about the mobility of components in the rigid portion of cell walls. The experiments were measured in a CP-based pseudo-3D format by measuring a series of 2D $^{13}$C–$^{13}$C correlation spectra with a variable z-filter time[81] of 0, 0.2, 1, 3, and 8 s. The relaxation data were fit using a single exponential decay function.

**Reporting summary**. Further information on research design is available in the Nature Research Reporting Summary linked to this article.

## Data availability

All NMR spectra and biochemical data that support the findings of this study are provided in the article, Supplementary Information. The processed topspin NMR datasets of cellular polysaccharides are available upon requests from the corresponding authors due to the large size of the files and lack of appropriate repository. Source data are provided with this paper.

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

## Acknowledgements

This work was supported by the National Institutes of Health (NIH) grant AI149289 to T.W. Preparation of isotopically labeled samples was supported by the Bagui Scholar Program Fund of Guangxi Zhuang Autonomous Region 2016A24 to C.J. A portion of this work was performed at the National High Magnetic Field Laboratory, which is supported by the National Science Foundation Cooperative Agreement No. DMR-1644779 and the State of Florida. The authors thank Françoise Baleux and Christelle Ganneau (Unité de Chimie des biomolecules, Institut Pasteur) for their help in the quantification of amino acids.

## Author contributions

A.C., L.D.F., and M.C.D.W. conducted the NMR experiments and analyzed the experimental data. W.F., P.W., C.J. and M.C.D.W. prepared the 13C, 15N-labeled fungal samples. T.F. and J.-P.L. provided the chemical analysis. J.-P. L., W.F., C.J. and T.W. designed and supervised the project. All authors contributed to paper writing.

## Competing interests

The authors declare no competing interests.
