## [Peer Review File · Nature Communications]

A Molecular Vision of Fungal Cell Wall Organization by
Functional Genomics and Solid-State NMRREVIEWER COMMENTS

Reviewer #1 (Remarks to the Author):

This work represents a follow up study of the seminal 2018 NMR paper (ref 14) to reconcile NMR and chemical evidence regarding the molecular and structural organization of the cell wall of a fungal pathogen *Aspergillus fumigatus*. In the current study four mutants depleted of major structural polysaccharides are analyzed by NMR, EM and GC/HPLC. In addition, an NMR analysis of alkali-soluble and insoluble fractions is presented with an emphasis on a & b-1,3 glycan, amino-acids/proteins and chitin.

The NMR results are clearly sensitive to the genetic mutations. In several cases however, it remains unclear what can reliably be concluded from the NMR data, in particular regarding quantitation of the different molecular species and their structural properties. The same applies to the protein/amino-acid analysis. Keeping in mind that different experimental conditions lead to different cell wall compositions, the paper needs revisions to support the general claims, also in relation to the 2018 work.

Specific comments are given below:

1. The authors generally distinguish between “rigid” and “mobile” species. However, there may be molecular species that exhibit medium time scale motion that is not captured by the NMR data shown. This aspect could complicate a quantitative analysis of the NMR data (vide infra). A possible means to investigate such issues would be to conduct experiments at variable temperatures and/or using frozen samples.
2. The authors compare in Fig. 1 and Fig. S1 NMR data obtained on parental (WT) and mutant strains. Especially Fig. S1 seems to suggest that the WT data are different to what was published in 2018(Fig. 1a) even though the expt. parameters seem to be largely the same. Please clarify.
3. In Fig. 1d, the authors estimate the polysaccharide composition using integrals of 2D NMR cross peaks. Unfortunately, I could not find the details of this procedure in the ms and how the significant variations in NMR line width (line 94) and the fact the T1 and T1rho relaxation times may vary were accounted for. Also, error bars are missing.

4. In Fig.2 the authors employ J-based NMR and short recycle delays to select for mobile residues (line 127). Yet, the data seems to exhibit significant variations in line width (0.4-0.8 ppm) – please explain. Also, it is well known that ^{13}C T1 times in solids are very sensitive to spin diffusion effects, especially in fully labeled samples and low MAS rates (see, e.g. Asami et al, JACS 2015) as in the current study. Again, the authors need to explain in more detail how they quantified in Fig. 2C the NMR data incl. time scales and provide error bars.

5. With these aspects in mind, it would also be useful to add a plot in which the NMR findings in Fig. 1 and 2 are correlated with the GC/HPLC data shown in Fig. S3. Again error bars seem to be missing in Fig. S3.

6. Analysis of AI samples in Fig. 3 confirms the 2018 results that α -1,3 glycan is closely associated with chitin (Line 198). Again, the authors seem to use NMR to quantify (in mole %) molecular species without giving error bars or further details how these numbers were obtained from NMR intensities for a given line width.

7. In Fig.4 the authors refer to “mobile proteins” and conclude that “valine is a prominent aa in the cell walls” (line 232) and find a “strong α -helicity of most residues except Tyr” (line 239). To the reviewer, the data presented in Fig. 4b are insufficient to draw such general conclusions: only 3-5 amino-acid types exhibit significant deviations from random coil values. In addition, there are no experimental data that would prove the existence of a polypeptide chain – e.g. via intra -or inter-residue NMR experiments. There are also no data that would support the claim in the abstract that valine may have “an unpredicted function ... in stabilizing macromolecular complexes”.

8. In fig. 5a.b ^{13}C T1 data are shown. As mentioned above, these parameters are (under the experimental conditions) dependent on dipolar couplings and spin diffusion effects. These effects can lead to T1 variations by an order of magnitude which is much larger than the changes seen here. Please comment

9. It is not entirely clear what can be reliably concluded from Figure 6; For example, the figure suggests an almost 50% reduction in cell wall thickness for Chitin def. variant while Fig. S10 contains error bars at least equal in size. Also, the location and structural properties of the molecular species penciled in for WT and mutant prep do not really differ – except of course for the species that has been deleted. Yet, line 330 claims that the current work “substantially” revises our understanding of the cell wall architecture. Please explain.

Reviewer #2 (Remarks to the Author):

Chakraborty et al., performed a comprehensive analysis of the cell wall of the major fungal pathogen *Aspergillus fumigatus*. This study builds on, partially confirms, and significantly extends the insights obtained in previous studies of the groups of Tuo Wang, specifically Kang et al., *Nat Commun.* 2018 Jul 16;9(1):2747, and of Jean-Paul Latge (who has a long history of fungal cell wall studies). In this study the authors made use of a set of different cell wall mutants that were created by the group of JP Latge over many years and which lack key components of the fungal cell wall of the investigated species, such as alpha-1,3-glucan, chitin, galactomannan (GM), galactosaminogalactan (GAG). These components are the major constituents, besides beta-1,3-glucan, of the cell wall of *A. fumigatus*. Compared to the previous study of the group of Tuo Wang (Kang et al., 2018), which analyzed the cell wall of wild type, this study highlights the impact of the lack of important individual cell wall components on the organization, dynamics, and hydration. Based on their results, the authors significantly refine the current model of the fungal cell wall organization, especially with regards to the structural relevance of individual components.

I must state that I have only a very limited understanding of NMR. I therefore cannot comment on the technical quality of NMR analysis and on whether the interpretation of the spectra is correct and appropriate. Importantly, due to the technical language describing the “polymer dynamics and hydration in the mutants” results section on page 13 and 14, I cannot comment on this part.

Major points:

It would be helpful if the authors explain or clearly define what they mean with the “mobile phase” of the cell wall.

Furthermore, the authors should explain in more detail what functional implications the different hydrated and hydrophobic domains may have.

In line 109, the authors state that no 1,6- and 1,4- linkages were seen due to the limited amount of such moieties. However, beta-1,4- and beta-1,6-glucan was found by this group previously in the cell wall of *A. fumigatus* by Kang et al. Why was it not found this time? Importantly, *A. fumigatus* was generally assumed and reported to have no beta-1,6-glucan in its cell wall.

The authors found, in agreement with their previous report (Henry et al., *mBio.* 2019 Feb 12;10(1):e02647-18) that chitin is drastically increased in the cell wall of the *krt4/ktr7* mutant which lacks galactomannan (GM) and which has a severe growth defect. According to the authors, GM is only in the mobile phases of the AS and AI fractions. Why is it that important for the cell wall integrity? Do

the authors think the increase of chitin is a direct consequence of the lack of GM or could there be an additional defect in the *krt4/ktr7* mutant which results in a compensatory increase of chitin? This might add another level of complexity to the conclusions (i.e., the effects seen in the “GM def.” mutant are not necessarily a consequence of the lack of GM).

Regarding Fig. 1 C: why are the chitin peaks seen in the α -1,3-glucan mutant, which are missing in the chitin mutant, also missing in the wild type even though the wild type has a similar or even higher chitin content than the α -1,3-glucan mutant(Fig. 1 D)?

In line 179, which “other mutants” are specifically meant? It is only “GM” mentioned after this. I assume they mean “In the GM mutant”?

In line 182/183 it is stated that the polysaccharide composition has ben “adjusted to compensate”. Are the authors sure that it is really compensation and, in some cases (throughout the manuscript), not rather a result of reduced production of the other components?

The authors mention discrepancies between the results of the NMR analysis and of previous studies that were based on chemical analysis (line 253). For example, it is pronounced that in the present study, in agreement with the previous study (Kang et al., 2018), NMR found high amounts of alpha-1,3-glucan in the AI fraction, but previous chemical analyses and the chemical analysis in this study did not (Sup. Table 5). Notably, a chemical analysis of the cell wall of a mutant that lack *fks1* (Dichtl et al., Mol Microbiol. 2015 Feb;95(3):458-71), the only beta-1,3-glucan synthase, also showed a glucose peak in the AI fraction. Could this also indicate the presence of alpha-1,3-glucan? The authors reason in the discussion that these differences could be linked to different media used in these studies. NMR was done with minimal medium, and the chemical analysis with Sabouraud and Brian medium (the exact recipe of these two are not described in the manuscript). I think it could be worth to analysis the cell wall in the used for the NMR analysis to clarify whether it is linked to the media as proposed by the authors or to the different technical approaches.

Reviewer #3 (Remarks to the Author):

In this report, Chakraborty et al. have put the powerful solid-state NMR methods this group developed for intact fungal cell walls to good use by investigating how the *Aspergillus fumigatus* pathogen reorganizes architecturally if one or more genes controlling the synthesis of its principal cell wall polysaccharides are deleted. This assessment benefits from judiciously chosen NMR acquisition methods

that provide estimates of relative composition of the chitin, various glucan, GM, and GAG species separately in the rigid core and mobile domain of the cell wall. The work also uses alkaline solubilization in a more comprehensive way than reported previously, conducting analyses separately for alkaline-soluble (AS) and alkaline-insoluble (AI) fractions of the fungal cell walls. Together, these strategies revealed significant compositional changes among the cell wall polysaccharides accompanying nearly every genetic modification. The unexpected finding of strong valine signals from possible polysaccharide-associated proteins in other than GM- and GAG-deficient mutants was intriguing (and should be investigated further) because it suggests roles for these polysaccharides in stabilizing proteins at the cell wall surface. Finally, measurements of polymer reorientation and water accessibility were used in an effort to evaluate how gene deletion alters the domain organization and hydrophobicity of the cell wall constituents.

Several specific concerns could benefit from further attention by the authors:

1. The failure to observe GAG using INADEQUATE C-13 NMR of AI and AS fractions (described on lines 229 ff) was problematic. Although it is plausible that GAG has become invisible due to unfavorable dynamic properties, the possibility of cell wall constituents that are observable by NEITHER CP-INADEQUATE nor short DP-INADEQUATE methods suggests a possible hole in spectroscopic analyses that are meant to be complete and reflective of the entire intact fungal cell. If I am misinterpreting this finding, then perhaps the presentation needs to be made with greater clarity.
2. The point noted above leads to one of several discrepancies noted by the authors between compositional analyses of particular glucan, GM, and GAG species in the AI and AS fractions that were carried out by NMR vs. chemical methods (253ff), but shouldn't explanations be proposed?
3. The claim that impaired biosynthesis of a cell wall polysaccharide causes structural responses such as increased rigidity and decreased water retention (396ff) looks correct, but it was difficult to tell whether the organizational arrangements in Figure 6 offer unique pictures of the resulting domain organization. Perhaps a more stepwise presentation of these models would make the proposals represent less of a logical jump for the reader.
4. Did the authors conduct measurements on biological or technical replicates to verify the consistency and correctness of their findings? These kinds of validation could be especially important for genetically modified materials.
5. A few errors in English usage jump out and could be corrected as follows: threatening instead of threatful (line 50); alteration rather than alternation (line 116); subtle differences rather than delicate differences (line 162); alteration rather than alternation (line 277); connect rather than pair? (line 322); involvement rather than evolvment
6. The very first sentence should specify that the number of people infected is two million per year, or another appropriate time period.
7. The term parental seems awkward. Should this be wild type?

Responses to Reviewers' Comments

Reviewer #1:

This work represents a follow up study of the seminal 2018 NMR paper (ref 14) to reconcile NMR and chemical evidence regarding the molecular and structural organization of the cell wall of a fungal pathogen *Aspergillus fumigatus*. In the current study four mutants depleted of major structural polysaccharides are analyzed by NMR, EM and GC/HPLC. In addition, an NMR analysis of alkali-soluble and insoluble fractions is presented with an emphasis on a & b-1,3 glycan, amino-acids/proteins and chitin.

The NMR results are clearly sensitive to the genetic mutations. In several cases however, it remains unclear what can reliably be concluded from the NMR data, in particular regarding quantitation of the different molecular species and their structural properties. The same applies to the protein/amino-acid analysis. Keeping in mind that different experimental conditions lead to different cell wall compositions, the paper needs revisions to support the general claims, also in relation to the 2018 work.

We thank the reviewer for the comments that are both critical and insightful. We have addressed all the points to improve the technical clarity and to better distinguish new findings from hypotheses and existing knowledge. A substantial amount of new data is also added to support the conclusions.

Specific comments are given below:

1. The authors generally distinguish between “rigid” and “mobile” species. However, there may be molecular species that exhibit medium time scale motion that is not captured by the NMR data shown. This aspect could complicate a quantitative analysis of the NMR data (vide infra). A possible means to investigate such issues would be to conduct experiments at variable temperatures and/or using frozen samples.

We fully agree with the reviewer about the potential complication by intermediate dynamics. Fortunately, we found that the combination of CP and short recycle delay DP could efficiently catch the vast majority of the molecules in these *A. fumigatus* samples. The results are now added as the new **Supplementary Fig. 1**. It also showed a consistent spectral pattern within a biologically favored range of temperature (280-298 K). We typically avoid approaching the freezing point to keep the sample as natural as possible. In addition, we have added the description of rigid and mobile molecules at the beginning of the Results section (**Lines 82-100**):

“Tailoring the solid-state NMR methods allowed us to selectively detect the rigid and mobile molecules as defined by their native dynamics in cell wall materials, with no relevance to covalent linkage patterns or their susceptibility to chemical extraction (for example, alkali treatment). It is quite common that a single type of polysaccharides could possibly have mobile domains that are present in the soft matrix and rigid phases that are physically packed with stiff molecules such as the cellulose microfibrils in plants and the chitin molecules in fungi.

The mobile phase represents those molecules with rapid ^{13}C - T_1 relaxation, which can survive through the short recycle delay used in ^{13}C direct polarization (DP). The rigid components described here refer to those polysaccharides that efficiently retain their dipolar couplings and thus can be detected using the dipolar-based ^1H - ^{13}C cross polarization (CP) method. These methods have been applied in solid-state NMR studies of carbohydrate-rich materials such as the cell walls of plants and algae as well as the biofilms and cell walls of fungi. In *A. fumigatus*, combining these two methods enables efficient detection of both mobile and rigid molecules at ambient temperature (Supplementary Fig. 1). This

physical vision differs from the classification accepted after chemical extraction and solubilization of the cell wall, where the alkali-soluble and water-insoluble molecules are recognized as amorphous polysaccharides as observed by electron microscopy (EM) whereas the alkali-insoluble fraction contains fibrillar molecules also seen by EM.”

2. The authors compare in Fig. 1 and Fig. S1 NMR data obtained on parental (WT) and mutant strains. Especially Fig. S1 seems to suggest that the WT data are different to what was published in 2018 (Fig. 1a) even though the expt. parameters seem to be largely the same. Please clarify.

The strain used in the 2018 study was a wild-type strain (RL 578). The origin of the parental strain used in this study (which is not a wild-type strain) has been now acknowledged. It is similar to a wild-type strain but with the deletion of the Ku80 gene, which is part of a heterodimer binding to DNA double strand breaks and is required for the non-homologous end joining DNA repair. Deletion of the Ku80 favors DNA integration at the right locus and makes the construction of mutants easier. It is widely used in the *Aspergillus* community. Its growth and pathobiological properties have been repeatedly shown to be similar to any wild-type gene. It is why it is used in this study.

This has been mentioned in the Introduction (**Lines 52-55**): “The parental strain is $\Delta\text{akuB}^{\text{KU80}}$, which is a widely used model strain resulting from the deletion of the KU80 gene to enhance homologous recombination. Based on this parental strain, four mutants, each of which selectively eliminates a major cell wall polysaccharide, were generated.”

We have also written two paragraphs in the Discussion section (**Lines 437-448**) to justify the differences observed between this study and the initial study of 2018.

“Earlier chemical analyses have shown that the composition varies between mycelium and conidium cell wall and the culture medium used, which could be at the origin of the discrepancies seen between our earlier ssNMR study and the present one. Previously, 1,6- and 1,4-linkages were identified in the β -glucans, with the former likely attributable to the branching points of β -1,3/1,6-glucan and the latter belonging to β -1,3/1,4-glucans. However, such signals were not detected in the current samples. The wild-type strain (RL 578) used in the previous study differs from the one for the current study. In addition, the fungal material used previously was obtained after 14 days of growth in unshaken conditions in a sucrose-based medium. Under these experimental conditions, the material recovered was somehow heterogenous with conidium and mycelium and autolyzed mycelium due to the long growth time. In the current study we use short culture times to recover actively growing mycelium and in a shaken condition to recover a homogenous mycelial pellet, which is a well-controlled system for analyzing the cell wall.”

3. In Fig. 1d, the authors estimate the polysaccharide composition using integrals of 2D NMR cross peaks. Unfortunately, I could not find the details of this procedure in the ms and how the significant variations in NMR line width (line 94) and the fact the T1 and T1rho relaxation times may vary were accounted for. Also, error bars are missing.

We have now added a new section “Estimation of carbohydrate composition using resolved NMR signals” in the **Supplementary Information** to detail the procedures used for obtaining peak integrals, calculating molecular composition, and propagating errors. The source data is provided as an excel file (also in response to the editorial request by the journal). As we are using peak volumes instead of

reading height, the perturbation by linewidth is minimal. Because most 2D methods are not precisely quantitative, we typically use “estimation” and avoid the word “quantification” in the manuscript.

The effect from ^{13}C - T_1 relaxation is small because the ^{13}C - T_1 time constants of cell wall molecules are on a different time scale (seconds) when compared with the CORD mixing time (53 ms). Even for the mobile molecules with a short ^{13}C - T_1 (e.g. an average of 1.3 s), only a negligible dephasing factor of 4% will be resulted. The dephasing factor will be much smaller for the rigid components with longer ^{13}C - T_1 . On the other hand, the T1rho relaxation involved in the initial magnetization created by the 1-ms CP serves as a dynamical filter to help the suppression of signals from mobile molecules as shown in the new **Supplementary Fig. 1**.

In addition, we added **Supplementary Table 1** and **Supplementary Figs. 1, 2, and 4** to show the experimental parameters, reproducibility, and distribution of linewidths, which should help the improvement of technical clarity.

4. In Fig.2 the authors employ J-based NMR and short recycle delays to select for mobile residues (line 127). Yet, the data seems to exhibit significant variations in line width (0.4-0.8 ppm) – please explain. Also, it is well known that ^{13}C T_1 times in solids are very sensitive to spin diffusion effects, especially in fully labeled samples and low MAS rates (see, e.g. Asami et al, JACS 2015) as in the current study. Again, the authors need to explain in more detail how they quantified in Fig. 2C the NMR data incl. time scales and provide error bars.

A detailed presentation of the linewidth distribution is now included in **Supplementary Fig. 2**. We have now added a discussion in (**Lines 150-161**) regarding the linewidth distribution of mobile molecules and the potential contributions from the heterogeneity in polymer dynamics and interactions in cellular samples. We have also discussed the contribution of spin exchange to the multiexponential feature of ^{13}C - T_1 relaxation in uniformly labeled materials and mentioned how it facilitated the differentiation of polymer nanodomains. A detailed explanation of the quantification procedures is provided in Supplementary Information. Error bars are added to **Fig. 2c**, as well as **Figs. 1c** and **2d**.

5. With these aspects in mind, it would also be useful to add a plot in which the NMR findings in Fig. 1 and 2 are correlated with the GC/HPLC data shown in Fig. S3.

We have now added the GC-HPLC data of a new sample (**Figs. 3d-f**), which was exactly the one analyzed by ssNMR. In contrast to ssNMR, the chemical partitioning leads to the destruction of the sample. Due to the limited amount to be analyzed, we could only perform one analysis. However, we did enough GC-HPLC-MS analyses of the cell wall of *A. fumigatus* in the last 10 years to be confident in the data presented. Originally, this is why we included the compositional analysis of cell wall produced in three different media to show that the amount of the different polysaccharides was largely similar in three media (even though there are slight discrepancies) and during repetitions. We feel now it is better to highlight only the chemical and enzymatical analysis of the same sample analyzed by ssNMR. Therefore, we removed the GC-HPLC chemical analyses of fungal cell walls from different media, which were only performed to verify that the analysis obtained by chemical extraction and hydrolysis was correct and in concordance with previous data for cell wall composition.

The purpose of conducting the chemical analysis on the NMR samples was to show we obtain consistent results between NMR and GC/HPLC data. The new results are now added as **Fig. 3d-f** and discussed in **Lines 268-287**, which has confirmed the presence (often underestimated during earlier chemical analysis) and minor contribution of α -1,3-glucan to the AI portion of *A. fumigatus* cell walls.

We also highlighted the interest of coupling biophysical, chemical and genomic approaches to analyze the cell wall (**Lines 419-425**): “These studies have shown the synergism of both chemical and biophysical methods. SsNMR has identified a prominent role of α -1,3 glucans in the cell wall structuration whereas chemical analysis has often missed the presence of this polysaccharide in the alkali-insoluble fraction. Similarly, the presence of β -1,3-glucans in the alkali-soluble fraction was underestimated. Chemical analysis may be more accurate to identify the presence of a polysaccharide in a very low concentration: this could be one of the reasons for not seeing the GAG signals in the AS fraction or chitin signals in the chitin-deficient mutants.”

6. Analysis of AI samples in Fig. 3 confirms the 2018 results that α -1,3 glycan is closely associated with chitin (Line 198). Again, the authors seem to use NMR to quantify (in mole %) molecular species without giving error bars or further details how these numbers were obtained from NMR intensities for a given line width.

Error bars are now added to **Fig. 1d, 2c, 2d**, as well as **Supplementary Tables 3-5**. The detailed procedures for the estimation of composition and the calculation of errors are now given as a new section in the **Supplementary Information**.

7. In Fig.4 the authors refer to “mobile proteins” and conclude that “valine is a prominent aa in the cell walls” (line 232) and find a “strong α -helicity of most residues except Tyr” (line 239). To the reviewer, the data presented in Fig. 4b are insufficient to draw such general conclusions: only 3-5 amino-acid types exhibit significant deviations from random coil values. In addition, there are no experimental data that would prove the existence of a polypeptide chain – e.g. via intra -or inter-residue NMR experiments. There are also no data that would support the claim in the abstract that valine may have “an unpredicted function ... in stabilizing macromolecular complexes”.

The evidence of valine’s putative function is its resistance to NaOH extraction, a procedure that was expected to only retain the covalently crosslinked polysaccharide core of the cell wall. The identification of valine in the rigid part of AI sample suggests its strong association with polysaccharides (**Fig. 4f and 4g**) even though we did not show (as rightly pointed by the reviewer) any covalent linkage of valine with the polysaccharides of the cell wall. This is further supported by the absence of rigid valine signals in the mutants lacking GAG and GM, which are polysaccharides associated with proteins (**Fig. 4e**).

In addition, we have shown by amino acid analysis after acid hydrolysis the presence of valine in the AI fraction which justifies our hypothesis of the strong association of valine with the cell wall. To agree with the comment of the reviewer we did soften the role of valine in cell wall construction in **Lines 315-318 and 334-342**, tuned down the statement of valine’s putative function in the abstract (**Line 10-12**), and modified the subtitle of the valine section (**Line 296**): “Valine, an amino acid associated with the rigid cell wall matrix.” We also avoid the use of “predominant” when describing the secondary structure.

8. In fig. 5a.b ^{13}C T1 data are shown. As mentioned above, these parameters are (under the experimental conditions) dependent on dipolar couplings and spin diffusion effects. These effects can lead to T1 variations by an order of magnitude which is much larger than the changes seen here. Please comment

We appreciate the very insightful comment. We have added a brief clarification in **Lines 352-355**. We fully agree that accurate measurement of ^{13}C - T_1 is not feasible under the current MAS and labeling conditions due to the heavily dipolar-coupled spin systems. The experimentally measured (or apparent) ^{13}C - T_1 could still serve as qualitative indicators of polymer dynamics in cell wall systems. This is benefited from 1) the coexistence of dynamically distinct nanodomains such as crystalline molecules (e.g. chitin and cellulose) and soft matrix meshes where many molecules can be considered as highly mobile or even partially solvated, 2) the large length scale of polymer networks, e.g. ~40 nm between two rigid cellulose microfibrils in plant primary cell walls and ~100-200 nm for fungal cell wall thickness, 3) the high degree of polymerization and the very large molecular size of each polysaccharide, and therefore 4) the very low population of different polymers on the intermolecular interface. ^{13}C spin exchange could not efficiently average out the ^{13}C - T_1 difference, which is also evidenced by the difficulty in observing intermolecular cross peaks, which can only be detected with very weak signals in a frozen sample (to immobilize the mobile/solvated molecules) and through an extremely long PDSM mixing of 1.5-3.0 s. Benefited from such unique features, ^{13}C - T_1 measurements have been extensively used as a way of evaluating the polymer dynamics and their spatial proximity to the crystalline cores in uniformly labeled cell walls.

9. It is not entirely clear what can be reliably concluded from Figure 6; For example, the figure suggests an almost 50% reduction in cell wall thickness for Chitin def. variant while Fig. S10 contains error bars at least equal in size. Also, the location and structural properties of the molecular species penciled in for WT and mutant prep do not really differ – except of course for the species that has been deleted. Yet, line 330 claims that the current work “substantially” revises our understanding of the cell wall architecture. Please explain.

The large error bar in the chitin synthase mutant is due to morphological differences in the mycelium not seen with other mutants (see reference 21). In this mutant, we see part of the mycelial cells which have balloon shape, while the other cells are normal which is the cause of the large error bar computed for this mutant. It is however clear from our TEM measurement that the cell walls of the mutants are thinner than the parental sample, and we agree with the reviewer that the cell wall thickness does not differ much between mutants.

We would like to thank the reviewer for pointing out the need to modify **Figure 6**. We have now included panel numbers to facilitate in-depth discussion of each sample. We have also updated the captions of Fig. 6 to indicate that the average thickness of each strain is used for the plotting. The expanded Discussion section (**Lines 399-475**) explains the major structural concepts of each figure panel in Fig.6. Briefly, **Fig. 6a** provides the first scheme of fungal cell wall through a combination of polymer dynamics and chemical digestibility. **Fig. 6b-e** present the first comparative vision of the compositional changes, average cell wall thickness, and the possible rearrangement of molecules for explaining the observed changes in dynamics and hydration. We hope with these changes, the structural concepts have become more accessible to our readers. Even though this representation remains hypothetical and should be further refined by follow-up experimental work, we feel that such a figure is of interest to present new propositions on the role of each polymer on cell wall biosynthesis and organization.

Reviewer #2:

Chakraborty et al., performed a comprehensive analysis of the cell wall of the major fungal pathogen *Aspergillus fumigatus*. This study builds on, partially confirms, and significantly extends the insights obtained in previous studies of the groups of Tuo Wang, specifically Kang et al., Nat Commun. 2018

Jul 16;9(1):2747, and of Jean-Paul Latge (who has a long history of fungal cell wall studies). In this study the authors made use of a set of different cell wall mutants that were created by the group of JP Latge over many years and which lack key components of the fungal cell wall of the investigated species, such as alpha-1,3-glucan, chitin, galactomannan (GM), galactosaminogalactan (GAG). These components are the major constituents, besides beta-1,3-glucan, of the cell wall of *A. fumigatus*. Compared to the previous study of the group of Tuo Wang (Kang et al., 2018), which analyzed the cell wall of wild type, this study highlights the impact of the lack of important individual cell wall components on the organization, dynamics, and hydration. Based on their results, the authors significantly refine the current model of the fungal cell wall organization, especially with regards to the structural relevance of individual components.

We thank the reviewer for the positive comments regarding our study. We hope the current study will provide a useful understanding of cell wall organization and a helpful strategy for investigating more fungal systems using the combination of NMR, biochemical, and genomics methods.

I must state that I have only a very limited understanding of NMR. I therefore cannot comment on the technical quality of NMR analysis and on whether the interpretation of the spectra is correct and appropriate. Importantly, due to the technical language describing the “polymer dynamics and hydration in the mutants” results section on page 13 and 14, I cannot comment on this part.

Major points:

It would be helpful if the authors explain or clearly define what they mean with the “mobile phase” of the cell wall.

Furthermore, the authors should explain in more detail what functional implications the different hydrated and hydrophobic domains may have.

Thanks for the helpful suggestion. We have now discussed the nature of mobile and rigid molecules in the first two paragraphs of the Results section (**Lines 81-101**). In these paragraphs, we have also clarified the differences between rigid, mobile vs fibrillar and amorphous which are the words classically used to define chemically identified cell wall components from TEM. In addition, we have also added brief examples to showcase the functional implications of different hydrated and dynamical domains in **Lines 347-349**.

In line 109, the authors state that no 1,6- and 1,4- linkages were seen due to the limited amount of such moieties. However, beta-1,4- and beta-1,6-glucan was found by this group previously in the cell wall of *A. fumigatus* by Kang et al. Why was it not found this time? Importantly, *A. fumigatus* was generally assumed and reported to have no beta-1,6-glucan in its cell wall.

Thanks for the insightful comments. We have now added a paragraph in the Discussion section (**Lines 437-448**) to explain the discrepancies between our previous study and the current one.

“Earlier chemical analyses have shown that the composition varies between mycelium and conidium cell wall and the culture medium used, which could be at the origin of the discrepancies seen between our earlier ssNMR study and the present one. Previously, 1,6- and 1,4-linkages were identified in the β -glucans, with the former likely attributable to the branching points of β -1,3/1,6-glucan and the latter belonging to β -1,3/1,4-glucans. However, such signals were not detected in the current samples. The wild-type strain (RL 578) used in the previous study differs from the one for the current study. In addition, the fungal material used previously was obtained after 14 days of growth in unshaken conditions in a sucrose-based medium. Under these experimental conditions, the material recovered

was somehow heterogenous with conidium and mycelium and autolyzed mycelium due to the long growth time. In the current study we use short culture times to recover actively growing mycelium and in a shaken condition to recover a homogenous mycelial pellet, which is a well-controlled system for analyzing the cell wall. ”

The authors found, in agreement with their previous report (Henry et al., mBio. 2019 Feb 12;10(1):e02647-18) that chitin is drastically increased in the cell wall of the *krt4/ktr7* mutant which lacks galactomannan (GM) and which has a severe growth defect. According to the authors, GM is only in the mobile phases of the AS and AI fractions. Why is it that important for the cell wall integrity? Do the authors think the increase of chitin is a direct consequence of the lack of GM or could there be an additional defect in the *krt4/ktr7* mutant which results in a compensatory increase of chitin? This might add another level of complexity to the conclusions (i.e., the effects seen in the “GM def.” mutant are not necessarily a consequence of the lack of GM).

The exact reason of the crucial effect of GM is not yet clear. We agreed with the reviewer that the effects seen in the GM deficient mutant are not necessarily a consequence of the lack of GM. In this mutant like in others cell wall mutants (and in other fungal species), a strong stress resulting from a gene deletion or antifungal treatment provoking a growth defect always induces an increase in chitin level which is responsible for protection. The comment of the reviewer is sound because at the same time the increase in chitin is protective, it can slow down fungal growth. Such hypothesis has never been really tested and the change in the cell wall structuration seen with ssNMR is a first hint to approach the problem of compensatory reactions in the fungal cell wall. We have now clarified this point in **Lines 143-144** and **212-213**.

Regarding Fig. 1 C: why are the chitin peaks seen in the α -1,3-glucan mutant, which are missing in the chitin mutant, also missing in the wild type even though the wild type has a similar or even higher chitin content than the α -1,3-glucan mutant(Fig. 1 D)?

We have fixed the issue by replotting **Fig. 1c** using more consistent contour lines, which better reflects the changes of polymer composition.

In line 179, which “other mutants” are specifically meant? It is only “GM” mentioned after this. I assume they mean “In the GM mutant”?

Thanks for pointing out the ambiguity. We have now rewritten that sentence as “Compared to the parental strain, the chitin-deficient mutant also altered the ratio of the three major monosaccharide residues in GAG (**Fig. 2d**).”

In line 182/183 it is stated that the polysaccharide composition has been “adjusted to compensate”. Are the authors sure that it is really compensation and, in some cases (throughout the manuscript), not rather a result of reduced production of the other components?

We thank the reviewer for this important point. It is true that there are no cell wall studies quantifying comparatively the percentage of the cell wall polysaccharides of the different mutants per total amount of the mycelium (in other words mg of total cell wall or mg of each polysaccharide per mg of total mycelium dry weight. The reduction in the size of the cell wall (**Supplementary Fig. 12**) suggests indeed that not only the respective percentage of the different polymers in the cell wall is modified but the overall synthesis of the cell wall component is quantitatively reduced. This point has now been included in the Discussion (**Lines 451-453**).

The authors mention discrepancies between the results of the NMR analysis and of previous studies that were based on chemical analysis (line 253). For example, it is pronounced that in the present study, in agreement with the previous study (Kang et al., 2018), NMR found high amounts of alpha-1,3-glucan in the AI fraction, but previous chemical analyses and the chemical analysis in this study did not (Sup. Table 5). Notably, a chemical analysis of the cell wall of a mutant that lack fks1 (Dichtl et al., Mol Microbiol. 2015 Feb;95(3):458-71), the only beta-1,3-glucan synthase, also showed a glucose peak in the AI fraction. Could this also indicate the presence of alpha-1,3-glucan? The authors reason in the discussion that these differences could be linked to different media used in these studies. NMR was done with minimal medium, and the chemical analysis with Sabouraud and Brian medium (the exact recipe of these two are not described in the manuscript). I think it could be worth to analysis the cell wall in the used for the NMR analysis to clarify whether it is linked to the media as proposed by the authors or to the different technical approaches.

These are extremely well taken point. Indeed, this study has pointed out that ssNMR has been very useful for correcting mistakes in early chemical analysis of cell wall composition. Similarly, and as shown in this manuscript too, a chemical approach can be very useful for identifying the presence of polymers (such as GAG) in low concentration.

We have now conducted a chemical analysis (**Figs 3d-f**) on the same samples characterized by ssNMR in this study. The new results presented in **Figs 3f** confirmed the minor contribution of a-1,3-glucan to the AI fraction. In the only beta-1,3-glucan synthase mutant constructed by the group of Wagener, the presence of a glucose peak in the AI fraction is probably alpha-1,3-glucan and we never thought about analyzing this peak with an alpha 1,3 glucanase. This is why these two methods are synergistically powerful as mentioned now in the Discussion (**Lines 419-425**).

The previous chemical data in Sabouraud and Brian medium were originally presented to confirm that the major components were always present when the fungus was grown in different media. Since we have now analyzed the mycelium produced for the ssNMR approach, we felt that these data were irrelevant, and we have now removed them in the revised manuscript. We have now clarified these points in **Lines 268-279**, together with a discussion of the very helpful reference (Mol Microbiol, 2015).

Reviewer #3:

In this report, Chakraborty et al. have put the powerful solid-state NMR methods this group developed for intact fungal cell walls to good use by investigating how the *Aspergillus fumigatus* pathogen reorganizes architecturally if one or more genes controlling the synthesis of its principal cell wall polysaccharides are deleted. This assessment benefits from judiciously chosen NMR acquisition methods that provide estimates of relative composition of the chitin, various glucan, GM, and GAG species separately in the rigid core and mobile domain of the cell wall. The work also uses alkaline solubilization in a more comprehensive way than reported previously, conducting analyses separately for alkaline-soluble (AS) and alkaline-insoluble (AI) fractions of the fungal cell walls. Together, these strategies revealed significant compositional changes among the cell wall polysaccharides accompanying nearly every genetic modification.

The unexpected finding of strong valine signals from possible polysaccharide-associated proteins in other than GM- and GAG-deficient mutants was intriguing (and should be investigated further) because it suggests roles for these polysaccharides in stabilizing proteins at the cell wall surface.

Finally, measurements of polymer reorientation and water accessibility were used in an effort to evaluate how gene deletion alters the domain organization and hydrophobicity of the cell wall constituents.

We really appreciate the encouraging comments of the technical and conceptual contributions of this study.

Several specific concerns could benefit from further attention by the authors:

1. The failure to observe GAG using INADEQUATE C-13 NMR of AI and AS fractions (described on lines 229 ff) was problematic. Although it is plausible that GAG has become invisible due to unfavorable dynamic properties, the possibility of cell wall constituents that are observable by NEITHER CP-INADEQUATE nor short DP-INADEQUATE methods suggests a possible hole in spectroscopic analyses that are meant to be complete and reflective of the entire intact fungal cell. If I am misinterpreting this finding, then perhaps the presentation needs to be made with greater clarity.

We have now provided new chemical data in **Lines 268-287** using the sample sample as the one used for our ssNMR measurement. The combination of CP and DP detection in 2D spectra should be sufficient for detecting the majority of molecules as shown by the new **Supplementary Fig. 1**. A small range of temperature has also been explored. The original batch of AS and AI NMR samples were used for the destructive analysis through chemical methods. We have prepared another fresh batch of sample but could not identify GAG signals. The yield of extraction was very low, giving less than 10 mg wet mass for the AS sample. GAG accounts for less than 10% of the AS material, therefore, we are limited by the sensitivity. We now pointed that the DNP might be a promising method for detecting lowly populated molecules in cellular systems and it is for sure a great opportunity for further investigating such molecules and understanding the solvation role of the NaOH treatment to hide the NMR signals. The insightful comments from the reviewer have inspired us to plan an experiment with pure GAG molecule purified from the culture filtrate without any chemical treatment (see reference 37) and then submitted to different chemical treatments to see their impact on the signals specific to GAG. These assays and the follow-up experiments will be time-consuming and might be more suitable as a separate study.

2. The point noted above leads to one of several discrepancies noted by the authors between compositional analyses of particular glucan, GM, and GAG species in the AI and AS fractions that were carried out by NMR vs. chemical methods (253ff), but shouldn't explanations be proposed?

Thanks for the helpful advice. We have added **Figs. 3d-f**. (together with the new description in **Lines 266-285**) to present the new chemical analysis conducted on the same AI and AS samples used for NMR measurements (but also destroyed these samples). The results are highly consistent with the NMR data, with α -1,3-glucan present as a minor fraction in AI but a dominant molecule in AS. The presence of mannan in both fractions is also confirmed. We have also discussed the synergism of chemical methods and ssNMR spectroscopy (**Lines 419-425**).

3. The claim that impaired biosynthesis of a cell wall polysaccharide causes structural responses such as increased rigidity and decreased water retention (396ff) looks correct, but it was difficult to tell whether the organizational arrangements in Figure 6 offer unique pictures of the resulting domain

organization. Perhaps a more stepwise presentation of these models would make the proposals represent less of a logical jump for the reader.

We fully agree that **Fig. 6** and the associated writing are insufficient for conveying the information. We have now reformatted **Fig. 6**, updated the caption, and rewritten the first and fourth paragraphs of the Discussion section (**Lines 399-417** and **Lines 450-475**). The first paragraph summarizes the new vision of parental fungal cell wall that combines the spectroscopic and biochemical evidence, and the fourth paragraph provides a stepwise explanation of the notable changes in each mutant. We have also rephrased the uncertain and hypothetical aspects of each scheme.

4. Did the authors conduct measurements on biological or technical replicates to verify the consistency and correctness of their findings? These kinds of validation could be especially important for genetically modified materials.

Thanks for the insightful advice. We have now prepared 3 additional batches of ^{13}C , ^{15}N -labeled samples for each strain (So in total, we have 4 replicates for each strain). The results are highly reproducible. We have now added a new figure (**Supplementary Fig. 2**) together with a description in the Results section (**Lines 100-101**) and Methods section (**Lines 529-530**).

5. A few errors in English usage jump out and could be corrected as follows: threatening instead of threatful (line 50); alteration rather than alternation (line 116); subtle differences rather than delicate differences (line 162); alteration rather than alternation (line 277); connect rather than pair? (line 322); involvement rather than evolvment

Thanks! We have corrected all the errors.

6. The very first sentence should specify that the number of people infected is two million per year, or another appropriate time period.

Thanks for pointing the ambiguity. We have now clarified that there are more than two million life-threatening infections each year.

7. The term parental seems awkward. Should this be wild type?

The parental strain is not wild-type because it has been genetically modified with a Ku80 deletion to enhance homologous recombination (da Silva Ferreira et al. *Eukaryot Cell* 5, 207-211, 2006) and is commonly used in the *Aspergillus* labs all around the world to make mutants. It has been verified that this Ku80 strain has the same pathogenicity as its wild-type strain from which it was constructed. The mutants were generated based on this “parental strain”. We have now clarified this point in the Introduction (**Lines 52-55**) and Methods section (**Line 518**).

Editorial Comments

* All Nature Communications manuscripts must include a “Data Availability” section after the Methods section but before the References. If any of the data can only be shared on request or are subject to restrictions, please specify the reasons and explain how, when, and by whom the data can be accessed.

Thanks for the instruction. We have now added the Data Availability section and the source data.

* To maximise the reproducibility of research data, we strongly encourage you to provide a file containing the raw data underlying the following types of display items:

- Any reported means/averages in box plots, bar charts, and tables
- Dot plots/scatter plots, especially when there are overlapping points
- Line graphs

The data should be provided in a single Excel file with data for each figure/table in a separate sheet, or in multiple labelled files within a zipped folder. Name this file or folder 'Source Data', and include a brief description in your cover letter. The "Data Availability" section should also include the statement "Source data are provided with this paper."

We have included an Excel file documenting the source data underlying Figs. 1d, 2c, 2d, 3d-f, and 4b, as well as Supplementary Figs 4, 10b-d, 11a-e, and 12. We have also included the statement to the "Data Availability".

* Please replace bar graphs with plots that feature information about the distribution of the underlying data. All data points should be shown for plots with a sample size less than 10. For larger sample sizes, please consider box-and-whisker or violin plots as alternatives. Measures of centrality, dispersion and/or error bars should be plotted and described in the figure legend.

We have replotted Fig. 5b as a box-and-whisker plot. For Fig. 5d, each molecule has a data size of less than 10; therefore, we kept all data points. Descriptions of errors are included in the legend.

REVIEWERS' COMMENTS

Reviewer #1 (Remarks to the Author):

The revised version of the paper is greatly improved, both in terms of technical detail as well as in reference to the general relevance of the reported data.

I recommend publication provided that the following (minor) issues are resolved:

- The new section "Estimation of carbohydrate composition..." states that only resolved NMR signals were used but also includes an approximation for the overlapping peaks of α -1,3-glucans. Please add a table detailing which resonances were used. Such a table would also clarify the missing parameter m_{poly} in the 1st equation.

-The introductory paragraph on solid state NMR (starting at line 42) suggests that Wang et al are the only group that have used solid-state NMR to obtain molecular and structural insight into fungal cell walls. For the non-expert reader, a more general introductory sentence citing previous work of other groups including those mentioned elsewhere in the paper seems appropriate.

Reviewer #2 (Remarks to the Author):

All my concerns were appropriately addressed in the revised manuscript. Thanks to the additional chemical analysis of the samples used for NMR, the issue with the discrepancies between the results from the present study and the previous studies could be well resolved. I think this study brings important new insights and significantly improves our understanding of the cell wall of *Aspergillus fumigatus*.

Reviewer #3 (Remarks to the Author):

The authors have presented thorough responses to my comments, including both adjusted text and new experiments. Their response to my concern regarding molecular species that might be missed by both

CP and DP NMR acquisition strategies (also raised by Reviewer 1) offers some additional confidence in their negative GAG finding and provides more explanation of 'rigid' and 'mobile' designations that are likely to be unfamiliar to non-specialists. Similarly, their explanation of the distinction between a parental and wild type strain (again also unclear to Reviewer 1) reaches out effectively to the non-specialist and provides a rationale for differences from their 2018 publication. Both the ssNMR measurements and the comparisons with chemical analyses have been made more rigorous through replication and/or analysis. Finally, their discussion of the structural schemes for fungal cell walls in parental vs. mutant strains is now more specific and considers the spectroscopic and biochemical evidence in a way that seems better integrated to me.

Responses to Reviewers' Comments

Reviewer #1

The revised version of the paper is greatly improved, both in terms of technical detail as well as in reference to the general relevance of the reported data.

We appreciate the insightful comments that guided the improvement of the manuscript.

I recommend publication provided that the following (minor) issues are resolved:

- The new section “Estimation of carbohydrate composition...” states that only resolved NMR signals were used but also includes an approximation for the overlapping peaks of α -1,3-glucans. Please add a table detailing which resonances were used. Such a table would also clarify the missing parameter m_{poly} in the 1st equation.

Thanks for the helpful advice. We have now added a new **Supplementary Table 5** to list the peaks (or cross peaks) used for the quantification of the mobile and rigid molecules. In the Supplementary Section “Estimation of carbohydrate composition using resolved NMR signals”, we have now clarified that “ m_{poly} is the total number of cell wall polysaccharides” and also pointed out that “The NMR peaks used for quantification are provided in Supplementary Table 5 and the Source Data file.”

-The introductory paragraph on solid state NMR (starting at line 42) suggests that Wang et al are the only group that have used solid-state NMR to obtain molecular and structural insight into fungal cell walls. For the non-expert reader, a more general introductory sentence citing previous work of other groups including those mentioned elsewhere in the paper seems appropriate.

We have rewritten this sentence to provide a broader view: “Recently, solid-state NMR spectroscopy has been employed to characterize the molecular architecture of cell walls and melanin deposition in multiple fungal species including *A. fumigatus*, *Cryptococcus neoformans*, and *Schizophyllum commune*¹⁵⁻²⁰.” The references from multiple NMR groups (which were cited in later sections previously) were also moved here.

Reviewer #2

All my concerns were appropriately addressed in the revised manuscript. Thanks to the additional chemical analysis of the samples used for NMR, the issue with the discrepancies between the results from the present study and the previous studies could be well resolved. I think this study brings important new insights and significantly improves our understanding of the cell wall of *Aspergillus fumigatus*.

Thanks for the encouraging comments. We hope the revised study is of broad interest to readers from both microbiology and spectroscopy.

Reviewer #3

The authors have presented thorough responses to my comments, including both adjusted text and new experiments. Their response to my concern regarding molecular species that might be missed by both

CP and DP NMR acquisition strategies (also raised by Reviewer 1) offers some additional confidence in their negative GAG finding and provides more explanation of 'rigid' and 'mobile' designations that are likely to be unfamiliar to non-specialists. Similarly, their explanation of the distinction between a parental and wild type strain (again also unclear to Reviewer 1) reaches out effectively to the non-specialist and provides a rationale for differences from their 2018 publication. Both the ssNMR measurements and the comparisons with chemical analyses have been made more rigorous through replication and/or analysis. Finally, their discussion of the structural schemes for fungal cell walls in parental vs. mutant strains is now more specific and considers the spectroscopic and biochemical evidence in a way that seems better integrated to me.

We really appreciate the helpful advices provided by the reviewer for improving the technical clarity of this manuscript.